# Convergence Guarantees
# for Adaptive Bayesian Quadrature Methods

**Motonobu Kanagawa[†,#,*] and Philipp Hennig[#]**
[†]EURECOM, Sophia Antipolis, France
[#]University of Tübingen and Max Planck Institute for Intelligent Systems, Tübingen, Germany
motonobu.kanagawa@eurecom.fr & philipp.hennig@uni-tuebingen.de

## Abstract

Adaptive Bayesian quadrature (ABQ) is a powerful approach to numerical integration that empirically compares favorably with Monte Carlo integration on problems of medium dimensionality (where non-adaptive quadrature is not competitive). Its key ingredient is an acquisition function that changes as a function of previously collected values of the integrand. While this adaptivity appears to be empirically powerful, it complicates analysis. Consequently, there are no theoretical guarantees so far for this class of methods. In this work, for a broad class of adaptive Bayesian quadrature methods, we prove consistency, deriving non-tight but informative convergence rates. To do so we introduce a new concept we call *weak adaptivity*. Our results identify a large and flexible class of adaptive Bayesian quadrature rules as consistent, within which practitioners can develop empirically efficient methods.

## 1 Introduction

Numerical integration, or quadrature/cubature, is a fundamental task in many areas of science and engineering. This includes machine learning and statistics, where such problems arise when computing marginals and conditionals in probabilistic inference problems. In particular in hierarchical Bayesian inference, quadrature is generally required for the computation of the *marginal likelihood*, the key quantity for model selection, and for *prediction*, for which latent variables are to be marginalized out.

To describe the problem, let $\Omega$ be a compact metric space, $\mu$ be a finite positive Borel measure on $\Omega$ (such as the Lebesgue measure on compact $\Omega \subset \mathbb{R}^d$) that playes the role of reference measure, $\pi : \Omega \to \mathbb{R}$ be a known density function, and $f : \Omega \to \mathbb{R}$ be an *integrand*, a known function such that the function value $f(x) \in \mathbb{R}$ can be obtained for any given query $x \in \Omega$. The task of quadrature is to numerically compute the integral (assumed to be intractable analytically)

$$\int f(x)\pi(x)d\mu(x).$$

This is done by evaluating the function values $f(x_1), \ldots, f(x_n)$ at design points $x_1, \ldots, x_n \in \Omega$ and using them to approximate $f$ and the integral. The points $x_1, \ldots, x_n$ should be "good" in the sense that $f(x_1), \ldots, f(x_n)$ provide useful information for computing the the integral.

Monte Carlo methods are the classic alternative, where $x_1, \ldots, x_n$ are randomly generated from a proposal distribution and the integral is approximated as $\sum_{i=1}^n w_i f(x_i)$, with $w_1, \ldots, w_n$ being importance weights. Such Monte Carlo estimators achieve the convergence rate of order $n^{-1/2}$ for $n$ the number of design points, under a mild condition that $f$ is a bounded function. This dimension-independent rate, and the mild condition about $f$, would be one of the reasons for the wide popularity and successes of Monte Carlo methods. However, as has been empirically known for practitioners

---

[*]Most of this work was done when MK was affiliated with University of Tübingen and MPI IS, Germany

and also theoretically investigated recently [3, 10], practical (i.e. Markov Chain) Monte Carlo can struggle in high dimensional integration, requiring a huge number of sample points to give a reliable estimate:[2] the curse of dimensionality appears in the constant term in front of the rate $n^{-1/2}$ [22, Sec. 2.5] [10, Thm. 2.1 and Sec. 3.4]. Thus, there has been a number of attempts on developing methods that work better than Monte Carlo for high dimensional integration, such as Quasi Monte Carlo methods [14].

Adaptive Bayesian quadrature (ABQ) is a recent approach from machine learning that actively, sequentially and deterministiclaly selects design points to adapt to the target integrand [29, 30, 16, 1, 9]. It is an extension of Bayesian quadrature (BQ) [28, 15, 8, 21], a probabilistic numerical method for quadrature that makes use of prior knowledge about the integrand, such as smoothness and structure, via a Gaussian process (GP) prior. Convergence rates of BQ methods take the form $n^{-s/d}$ if the integrand $f$ is $s$-times differentiable, or of the form $\exp(-Cn^{1/d})$ for some constant $C > 0$ if $f$ is infinitely smooth [8, 20]. While the rates can be faster than Monte Carlo, the dimension $d$ of the ambient space now appears in the rate, meaning that BQ also suffers from the curse of dimensionality.

ABQ has been developed to improve upon such vanilla BQ methods. One drawback of vanilla BQ is that the Gaussian process model prevents the use of certain kinds of relevant knowledge about the integrand, such as it being *positive* (or non-negative), because they cannot be encoded in a Gaussian distribution. Positive integrands are ubiquitous in machine learning and statistics, where integration tasks emerge in the marginalization and conditioning of probability density functions, which are positive by definition. In ABQ such prior knowledge is modelled by describing the integrand as given by a certain *transformation* (or warping) of a GP — for instance, an exponentiated GP [30, 29, 9] or a squared GP [16]. ABQ methods with such transformations have empirically been shown to improve upon both standard BQ and Monte Carlo, leading to state-of-the-art wall-clock time performance on problems of medium dimensionality.

If the transformation is nonlinear, as in the examples above, the transformed GP no longer allows an analytic expression for its posterior process, and thus approximations are used to obtain a tractable acquisition function. In contrast to the posterior covariance of GPs, these acquisition functions then become dependent on previous observations, making the algorithm adaptive. This twist seems to be critical for ABQ methods' superior empirical performance, but it complicates analysis. Thus, there has been no theoretical guarantee for their convergence, rendering them heuristics in practice. This is problematic since integration is usually an intermediate computational step in a larger system, and thus must be reliable. This paper provides the first convergence analysis for ABQ methods.

In Sec. 2 we review ABQ methods, and formulate a generic class of acquisition functions that cover those of [16, 1, 2, 9]. Our convergence analysis is done for this class. We also derive an upper-bound on the quadrature error using a transformed integrand, which is applicable to any design points and given in terms of the GP posterior variance (Prop. 2.1). In Sec. 3, we establish a connection between ABQ and certain *weak greedy algorithms* (Thm. 3.3). This is based on a new result that the scaled GP posterior variance can be interpreted in terms of a certain projection in a Hilbert space (Lemma 3.1). Using this connection, we derive convergence rates of ABQ methods in Sec. 4. For ease of the reader, we present a high-level overview of the proof structure in Fig. 1.

The key to our analysis is a relatively general notion for active exploration that we term *weak adaptivity*. An ABQ method that satisfies weak adaptivity (and a few additional technical constraints) is consistent, and the conceptual space of weakly adaptive BQ methods is large and flexible. We hope that our results spark a practical interest in the design of empirically efficient acquisition functions, to extend the reach of quadrature to problems of higher and higher dimensionality.

**Related Work.**   For standard BQ methods, and the corresponding *kernel quadrature* rules, convergence properties have been studied extensively [e.g. 7, 19, 4, 40, 21, 11, 8, 27, 20]. Some of these works theoretically analyze methods that deterministically generate design points [12, 5, 17, 7, 11]. These methods are, however, *not* adaptive, as design points are generated independently to the function values of the target integrand.

Our analysis is technically related to the work by Santin and Haasdonk [34], which analyzed the so-called *P-greedy algorithm*, an algorithm to sequentially obtain design points using the GP posterior

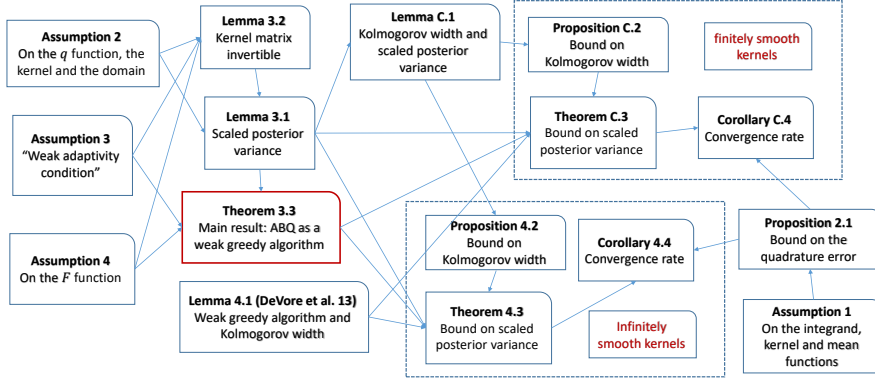

Figure 1: Relationships between the various auxiliary results and how they yield the main results.

variance as an acquisition function. Our results can be regarded as a generalization of their result so that the acquisition function can include i) a scaling and a transformation of the GP posterior variance and ii) a data-dependent term that takes care of adaptation; see (4) for details.

Adaptive methods have also been theoretically studied in the information-based complexity literature [23, 24, 25, 26]. The key result is that optimal points for quadrature can be obtained *without* observing actual function values, *if* the hypothesis class of functions is *symmetric and convex* (e.g. the unit ball in a Hilbert space): in this case adaptation does *not* help improve the performance. On the other hand, it the hypothesis class is either *asymmetric* or *nonconvex*, then adaptation may be helpful. For instance, a class of positive functions is assymetric because only one of $f$ or $-f$ can be positive. These results thus support the choice of acquisition functions of existing ABQ methods, where the adaptivity to function values is motivated by modeling the positivity of the integrand.

**Notation.** $\mathbb{N}$ denotes the set of positive integers, $\mathbb{R}$ the real line, and $\mathbb{R}^d$ the $d$-dimensional Euclidean space for $d \in \mathbb{N}$. $L_p(\Omega)$ for $1 \leq p < \infty$ is the Banach space of $p$-integrable functions, and $L_\infty(\Omega)$ is that of essentially bounded functions.

## 2 Adaptive Bayesian Quadrature (ABQ)

We describe here ABQ methods, and present a generic form of acquisition functions that we analyze. We also derive an upper-bound on the quadrature error using a transformed integrand in terms of the GP posterior variance, motivating our analysis in the later sections. Throughout the paper we assume that the domain $\Omega$ is a compact metric space and $\mu$ is a finite positive Borel measure on $\Omega$.

### 2.1 Bayesian Quadrature with Transformation

ABQ methods deal with an integrand $f$ that is a priori known to satisfy a certain constraint, for example $f(x) > 0 \ \forall x \in \Omega$. Such a constraint is modeled by considering a certain transformation $T : \mathbb{R} \to \mathbb{R}$, and assuming that there exists a latent function $g : \Omega \to \mathbb{R}$ such that the integrand $f$ is given as the transformation of $g$, i.e., $f(x) = T(g(x)), x \in \Omega$. Examples of $T$ for modeling the positivity include i) the square transformation $T(y) = \alpha + \frac{1}{2}y^2$, where $\alpha > 0$ is a small constant such that $0 < \alpha < \inf_{x \in \Omega} f(x)$, assuming that $f$ is bounded away from 0 [16]; and ii) the exponential transformation $T(y) = \exp(y)$ [30, 29, 9]. Note that the identity map $T(y) = y$ recovers standard Bayesian quadrature (BQ) methods [28, 15, 7, 21]. To model the latent function $g$, a Gaussian process (GP) prior [32] is placed over $g$:

$$g \sim \mathcal{GP}(m, k) \tag{1}$$

where $m : \Omega \to \mathbb{R}$ is a mean function and $k : \Omega \times \Omega$ is a covariance kernel. Both $m$ and $k$ should be chosen to capture as much prior knowledge or belief about $g$ (or its transformation $f$) as possible, such as smoothness and correlation structure; see e.g. [32, Chap. 4].

Assume that a set of points $X_n := \{x_1, \ldots, x_n\} \subset \Omega$ are given, such that the kernel matrix $K_n := (k(x_i, x_j))_{i,j=1}^n \subset \mathbb{R}^{n \times n}$ is invertible. Given the function values $f(x_1), \ldots, f(x_n)$, define

$g_i(x) := z_i \in \mathbb{R}$ such that $T(z_i) = f(x_i)$ for $i = 1, \ldots, n$. Treating $g(x_1), \ldots, g(x_n)$ as "observed data without noise," the posterior distribution of $g$ under the GP prior (1) is again given as a GP

$$g|(x_i, g(x_i))_{i=1}^n \sim \mathcal{GP}(m_{g,X_n}, k_{X_n}),$$

where $m_{g,X_n} : \Omega \to \mathbb{R}$ is the posterior mean function and $k_{X_n} : \Omega \times \Omega \to \mathbb{R}$ is the posterior covariance kernel given by (see e.g. [32])

$$
\begin{align}
m_{g,X_n}(x) &:= m(x) + \boldsymbol{k}_n(x)^\top K_n^{-1}(\boldsymbol{g}_n - \boldsymbol{m}_n), \tag{2} \\
k_{X_n}(x, x') &:= k(x, x') - \boldsymbol{k}_n(x)^\top K_n^{-1} \boldsymbol{k}_n(x'), \tag{3}
\end{align}
$$

where $\boldsymbol{k}_n(x) := (k(x, x_1), \ldots, k(x, x_n))^\top \in \mathbb{R}^n$, $\boldsymbol{g}_n := (g(x_1), \ldots, g(x_n))^\top \in \mathbb{R}^n$ and $\boldsymbol{m}_n = (m(x_1), \ldots, m(x_n))^\top \in \mathbb{R}^n$. Then a quadrature estimate[3] for the integral $\int f(x)\pi(x)d\mu(x)$ is given as the integral $\int T(m_{g,X_n}(x))\pi(x)d\mu(x)$ of the transformed posterior mean function $T(m_{g,X_n})$, or as the integral of the posterior expectation of the transformation $\int \mathbb{E}_{\acute{g}} T(\acute{g}(x))\pi(x)d\mu(x)$, where $\acute{g} \sim \mathcal{GP}(m_{g,X_n}, k_{X_n})$ is the posterior GP. The posterior covariance for $\int f(x)\pi(x)$ is given similarly; see [9, 16] for details.

## 2.2 A Generic Form of Acquisition Functions

The key remaining question is how to select good design points $x_1, \ldots, x_n \in \Omega$. ABQ methods sequentially and deterministically generate $x_1, \ldots, x_n$ using an *acquisition function*. Many of the acquisition functions can be formulated in the following generic form:

$$x_{\ell+1} \in \arg\max_{x \in \Omega} a_\ell(x), \quad \text{where} \quad a_\ell(x) = F\left(q^2(x)k_{X_\ell}(x, x)\right)b_\ell(x), \quad (\ell = 0, 1, \ldots, n-1) \tag{4}$$

where $k_{X_0}(x, x) := k(x, x)$, $F : [0, \infty) \to [0, \infty)$ is an increasing function such that $F(0) = 0$, $q : \Omega \to (0, \infty)$ and $b_\ell : \Omega \to \mathbb{R}$ is a function that may change at each iteration $\ell$. e.g., it may depend on the function values $f(x_1), \ldots, f(x_\ell)$ of the target integrand $f$. Intuitively, $b_\ell(x)$ is a data-dependent term that makes the point selection adaptive to the target integrand, $q(x)$ may be seen as a proposal density in importance sampling, and $F$ determines the balance between the uncertainty sampling part $q^2(x)k_{X_\ell}(x, x)$ and the adaptation term $b_\ell(x)$. We analyse ABQ with this generic form (4), aiming for results with wide applicability. Here are some representative choices.

**Warped Sequential Active Bayesian Integration (WSABI) [16]:** Gunter et al. [16] employ the square transformation $f(x) = T(g(x)) = \alpha + \frac{1}{2}g^2(x)$ with two acquisition functions: i) WSABI-L [16, Eq. 15], which is based on linearization of $T$ and recovered with $F(y) = y$, $q(x) = \pi(x)$ and $b_\ell(x) = m_{g,X_\ell}^2(x)$; and ii) WSABI-M [16, Eq. 14], the one based on moment matching given by $F(y) = y$, $q(x) = \pi(x)$ and $b_\ell(x) = \frac{1}{2}k_{X_\ell}(x, x) + m_{g,X_\ell}^2(x)$.

**Moment-Matched Log-Transformation (MMLT) [9]:** Chai and Garnett [9, 3rd raw in Table 1] use the exponential transformation $f(x) = T(g(x)) = \exp(g(x))$ with the acquisition function given by $F(y) = \exp(y) - 1$, $q(x) = 1$ and $b_\ell(x) = \exp(k_{X_\ell}(x, x) + 2m_{g,X_\ell}(x))$.

**Variational Bayesian Monte Carlo (VBMC) [1, 2]:** Acerbi [2, Eq. 2] uses the identity $f(x) = T(g(x)) = g(x)$ with the acquisition function given by $F(y) = y^{\delta_1}$, $q(x) = 1$ and $b_\ell(x) = \pi_\ell^{\delta_2}(x)\exp(\delta_3 m_{g,X_\ell}(x))$, where $\pi_\ell$ is the variational posterior at the $\ell$-th iteration and $\delta_1, \delta_2, \delta_3 \geq 0$ are constants: setting $\delta_1 = \delta_2 = \delta_3 = 1$ recovers the original acquisition function [1, Eq. 9]. Acerbi [1, Sec. 2.1] considers an integrand $f$ that is defined as the logarithm of a joint density, while $\pi$ is an intractable posterior that is gradually approximated by the variational posteriors $\pi_\ell$.

For the WSABI and MMLT, the acquisition function (4) is obtained by a certain approximation for the posterior variance of the integral $\int f(x)\pi(x)d\mu(x) = \int T(g(x))\pi(x)d\mu(x)$; thus this is a form of *uncertainty sampling*. Such an approximation is needed because the posterior variance of the integral is not available in closed form, due to the nonlinear transformation $T$. The resulting acquisition function includes the data-dependent term $b_\ell(x)$, which encourages exploration in regions where the value of $g(x)$ is expected to be large. This makes ABQ methods adaptive to the target integrand. Alas, it also complicates analysis. Thus there has been no convergence guarantee for these ABQ methods; which is what we aim to remedy in this paper.

## 2.3 Bounding the Quadrature Error with Transformation

Our first result, which may be of independent interest, is an upper-bound on the error for the quadrature estimate $\int T(m_{g,X_n}(x))\pi(x)d\mu(x)$ based on a transformation described in Sec. 2.1. It is applicable to any point set $X_n = \{x_1, \ldots, x_n\}$, and the bound is given in terms of the posterior variance $k_{X_n}(x, x)$. This gives us a motivation to study the behavior of this quantity for $x_1, \ldots, x_n$ generated by ABQ (4) in the later sections. Note that the essentially same bound holds for the other estimator $\int \mathbb{E}_{\acute{g}} T(\acute{g}(x))\pi(x)d\mu(x)$ with $\acute{g} \sim \mathcal{GP}(m_{g,X_n}, k_{X_n})$, which we describe in Appendix A.2.

To state the result, we need to introduce the Reproducing Kernel Hilbert Space (RKHS) of the covariance kernel $k$ of the GP prior. See e.g. [35, 36] for details of RKHS's, and [6, 18] for discussions of their close but subtle relation to the GP notion. Let $\mathcal{H}_k$ be the RKHS associated with the covariance kernel $k$ of the GP prior (1), with $\langle \cdot, \cdot \rangle_{\mathcal{H}_k}$ and $\| \cdot \|_{\mathcal{H}_k}$ being its inner-product and norm, respectively. $\mathcal{H}_k$ is a Hilbert space consisting of functions on $\Omega$, such that i) $k(\cdot, x) \in \mathcal{H}_k$ for all $x \in \Omega$, and ii) $h(x) = \langle k(\cdot, x), h \rangle_{\mathcal{H}_k}$ for all $h \in \mathcal{H}_k$ and $x \in \Omega$ (the reproducing property), where $k(\cdot, x)$ denotes the function of the first argument such that $y \to k(y, x)$, with $x$ being fixed. As a set of functions, $\mathcal{H}_k$ is given as the closure of the linear span of such functions $k(\cdot, x)$, i.e., $\mathcal{H}_k = \overline{\text{span} \{k(\cdot, x) \mid x \in \Omega\}}$, meaning that any $h \in \mathcal{H}_k$ can be written as $h = \sum_{i=1}^\infty \alpha_i k(\cdot, y_i)$ for some $(\alpha_i)_{i=1}^\infty \subset \mathbb{R}$ and $(y_i)_{i=1}^\infty \subset \Omega$ such that $\|h\|_{\mathcal{H}_k}^2 = \sum_{i,j=1}^\infty \alpha_i \alpha_j k(y_i, y_j) < \infty$. We are now ready to state our assumption:

**Assumption 1.** $T : \mathbb{R} \to \mathbb{R}$ *is continuously differentiable. For* $f : \Omega \to \mathbb{R}$, *there exists* $g : \Omega \to \mathbb{R}$ *such that* $f(x) = T(g(x)), x \in \Omega$ *and that* $\tilde{g} := g - m \in \mathcal{H}_k$. *It holds that* $\|k\|_{L_\infty(\Omega)} := \sup_{x \in \Omega} k(x, x) < \infty$ *and* $\|m\|_{L_\infty(\Omega)} := \sup_{x \in \Omega} |m(x)| < \infty$.

The assumption $\tilde{g} := g - m \in \mathcal{H}_k$ is common in theoretical analysis of standard BQ methods, where $T(y) = y$ and $m = 0$ [see e.g. 7, 40, 8, and references therein]. This assumption may be weakened by using proof techniques developed for standard BQ in the misspecifid setting [19, 20], but we leave it for a future work. The other conditions on $T$, $k$ and $m$ are weak.

**Proposition 2.1.** *(proof in Appendix A.1) Let* $\Omega$ *be a compact metric space,* $X_n = \{x_1, \ldots, x_n\} \subset \Omega$ *be such that the kernel matrix* $K_n = (k(x_i, x_j))_{i,j=1}^n \in \mathbb{R}^{n \times n}$ *is invertible, and* $\pi : \Omega \to [0, \infty)$ *and* $q : \Omega \to [0, \infty)$ *be continuous functions such that* $C_{\pi/q} := \int_\Omega \pi(x)/q(x)d\mu(x) < \infty$. *Suppose that Assumption 1 is satisfied. Then there exists a constant* $C_{\tilde{g},m,k,T}$ *depending only on* $\tilde{g}$, $m$, $k$ *and* $T$ *such that*

$$\left| \int f(x)\pi(x)d\mu(x) - \int T\left(m_{g,X_n}(x)\right)\pi(x)d\mu(x) \right| \le C_{\tilde{g},m,k,T} C_{\pi/q} \|\tilde{g}\|_{\mathcal{H}_k} \sup_{x \in \Omega} q(x)\sqrt{k_{X_n}(x, x)}.$$

Prop. 2.1 shows that to establish convergence guarantees for ABQ methods, it is sufficient to analyze the convergence behavior of the quantity $\sup_{x \in \Omega} q(x)\sqrt{k_{X_n}(x, x)}$ for points $X_n = \{x_1, \ldots, x_n\}$ generated from (4). This is what we focus on in the remainder.

## 3 Connections to Weak Greedy Algorithms in Hilbert Spaces

To analyze the quantity $\sup_{x \in \Omega} q(x)\sqrt{k_{X_n}(x, x)}$ for points $X_n = \{x_1, \ldots, x_n\}$ generated from ABQ (4), we show here that the ABQ can be interpreted as a certain *weak greedy algorithm* studied by DeVore et al. [13]. To describe this, let $\mathcal{H}$ be a (generic) Hilbert space and $\mathcal{C} \subset \mathcal{H}$ be a compact subset. To define some notation, let $h_1, \ldots, h_n \in \mathcal{C}$ be given. Denote by $S_n := \text{span}(h_1, \ldots, h_n) = \{\sum_{i=1}^n \alpha_i h_i \mid \alpha_1, \ldots, \alpha_n \in \mathbb{R}\} \subset \mathcal{H}$ the linear subspace spanned by $h_1, \ldots, h_n$. For a given $h \in \mathcal{C}$, let $\text{dist}(h, S_n)$ be the distance between $h$ and $S_n$ defined by

$$\text{dist}(h, S_n) := \inf_{g \in S_n} \|h - g\|_{\mathcal{H}} = \inf_{\alpha_1, \ldots, \alpha_n \in \mathbb{R}} \|h - \sum_{i=1}^n \alpha_i h_i\|_{\mathcal{H}},$$

where $\| \cdot \|_{\mathcal{H}}$ denotes the norm of $\mathcal{H}$. Geometrically, this is the distance between $h$ and its orthogonal projection onto the subspace $S_n$. The task considered in [13] is to select $h_1, \ldots, h_n \in \mathcal{C}$ such that the worst case error in $\mathcal{C}$ defined by

$$e_n(\mathcal{C}) := \sup_{h \in \mathcal{C}} \text{dist}(h, S_n) \tag{5}$$

becomes as small as possible: $h_1, \ldots, h_n \in \mathcal{C}$ are to be chosen to approximate well the set $\mathcal{C}$.

The following weak greedy algorithm is considered in DeVore et al. [13]. Let $\gamma$ be a constant such that $0 < \gamma \leq 1$, and let $n \in \mathbb{N}$. First select $h_1 \in \mathcal{C}$ such that $\|h_1\|_{\mathcal{H}} \geq \gamma \sup_{h \in \mathcal{C}} \|h\|_{\mathcal{H}}$. For $\ell = 1, \ldots n - 1$, suppose that $h_1, \ldots, h_\ell$ have already been generated, and let $S_\ell = \text{span}(h_1, \ldots, h_\ell)$. Then select a next element $h_{\ell+1} \in \mathcal{C}$ such that

$$\text{dist}(h_{\ell+1}, S_\ell) \geq \gamma \sup_{h \in \mathcal{C}} \text{dist}(h, S_\ell), \quad (\ell = 1, \ldots, n-1). \tag{6}$$

In this paper we refer to such $h_1, \ldots, h_n$ as *a $\gamma$-weak greedy approximation of $\mathcal{C}$ in $\mathcal{H}$* because, $\gamma = 1$ recovers the standard greedy algorithm, while $\gamma < 1$ weakens the "greediness" of this rule. DeVore et al. [13] derived convergence rates of the worst case error (5) as $n \to \infty$ for $h_1, \ldots, h_n$ generated from this weak greedy algorithm.

**Weak Greedy Algorithms in the RKHS.** To establish a connection to ABQ, we formulate the weak greedy algorithm in an RKHS. Let $\mathcal{H}_k$ be the RKHS of the covariance kernel $k$ as in Sec. 2.3, and $q(x)$ be the function in (4). We define a subset $\mathcal{C}_{k,q} \subset \mathcal{H}_k$ by

$$\mathcal{C}_{k,q} := \{q(x)k(\cdot, x) \mid x \in \Omega\} \subset \mathcal{H}_k.$$

Note that $C_{k,q}$ is the image of the mapping $x \to q(x)k(\cdot, x)$ with $\Omega$ being the domain. Therefore $\mathcal{C}_{k,q}$ is compact, if $k$ and $q$ are continuous and $\Omega$ is compact; this is because in this case the mapping $x \to q(x)k(\cdot, x)$ becomes continuous, and in general the image of a continuous mapping from a compact domain is compact. Thus, we make the following assumption:

**Assumption 2.** $\Omega$ *is a compact metric space, $q : \Omega \to \mathbb{R}$ is continuous with $q(x) > 0$ for all $x \in \Omega$, and $k : \Omega \times \Omega \to \mathbb{R}$ is continuous.*

The following simple lemma establishes a key connection between weak greedy algorithms and ABQ. (Note that the the result for the case $q(x) = 1$ is well known in the literature, and the novelty lies in that we allow for $q(x)$ to be non-constant.) For a geometric interpretation of (7) in terms of projections, see Fig.2 in Appendix B.1.

**Lemma 3.1.** *(proof in Appendix B.1) Let $x_1, \ldots, x_n \in \Omega$ be such that the kernel matrix $K_n = (k(x_i, x_j))_{i,j=1}^n \in \mathbb{R}^{n \times n}$ is invertible. Define $h_x := q(x)k(\cdot, x)$ for any $x \in \mathcal{X}$, and let $S_n := \text{span}(h_{x_1}, \ldots, h_{x_n}) \subset \mathcal{H}_k$. Assume that $q(x) > 0$ holds for all $x \in \Omega$. Then for all $x \in \Omega$ we have*

$$q^2(x)k_{X_n}(x, x) = \text{dist}^2(h_x, S_n), \tag{7}$$

*where $k_{X_n}(x, x)$ is the GP posterior variance function given by (3). Moreover, we have*

$$e_n(\mathcal{C}_{k,q}) = \sup_{x \in \Omega} q(x)\sqrt{k_{X_n}(x, x)}, \tag{8}$$

*where $e_n(\mathcal{C}_{k,q})$ is the worst case error defined by (5) with $\mathcal{C} := \mathcal{C}_{k,q}$ and $S_n$ defined here.*

Lemma 3.1 (8) suggests that we can analyze the convergence properties of $\sup_{x \in \Omega} q(x)\sqrt{k_{X_n}(x, x)}$ for $X_n = \{x_1, \ldots, x_n\}$ generated from the ABQ rule (4) by analyzing those of the worst case error $e_n(\mathcal{C}_{k,q})$ for the corresponding elements $h_{x_1}, \ldots, h_{x_n}$, where $h_{x_i} := q(x_i)k(\cdot, x_i)$.

**Adaptive Bayesian Quadrature as a Weak Greedy Algorithm.** We now show that the ABQ (4) gives a weak greedy approximation of the compact set $\mathcal{C}_{k,q}$ in the RKHS $\mathcal{H}_k$ in the sense of (6). We summarize required conditions in Assumptions 3 and 4. As mentioned in Sec. 1, Assumption 3 is the crucial one: its implications for certain specific ABQ methods will be discussed in Sec. 4.2.

**Assumption 3 (Weak Adaptivity Condition).** *There are constants $C_L, C_U > 0$ such that $C_L < b_\ell(x) < C_U$ holds for all $x \in \Omega$ and for all $\ell \in \mathbb{N} \cup \{0\}$.*

Intuitively, this condition enforces ABQ to not overly focus on a specific local region in $\Omega$ and to explore the entire domain $\Omega$. For instance, consider the following two situations where Assumption 3 does not hod.: (a) $b_\ell(x) \to +0$ as $\ell \to \infty$ for some local region $x \in A \subset \Omega$, while $b_\ell(x)$ remains bounded from blow for $x \in \Omega \backslash A$; (b) $b_\ell(x) \to +\infty$ as $\ell \to \infty$ for some local region $x \in B \subset \Omega$, while $b_\ell(x)$ remains bounded from above for $x \in \Omega \backslash B$. In case (a), ABQ will not allocate any points to this region $A$ at all, after a finite number of iterations. Thus, the information about the integrand $f$

on this region $A$ will not be obtained after a finite number of evaluations, which makes it difficult to guarantee the consistency of quadrature, unless $f$ has a finite degree of freedom on $A$. Similarly, in case (b), ABQ will generate points only in the region $B$ and no point in the rest of the region $\Omega \backslash B$, after a finite number of iterations. Assumption 3 prevents such problematic situations to occur.

**Assumption 4.** *$F : [0, \infty) \rightarrow [0, \infty)$ is increasing and continuous, and $F(0) = 0$. For any $0 < c \leq 1$, there is a constant $0 < \psi(c) \leq 1$ such that $F^{-1}(cy) \geq \psi(c)F^{-1}(y)$ holds for all $y \geq 0$.*

For instance, if $F(y) = y^\delta$ for $\delta > 0$ then $F^{-1}(y) = y^{1/\delta}$ and thus we have $\psi(c) = c^{1/\delta}$ for $0 < c \leq 1$; $\delta = 1$ is the case for the WSABI [16], and $\delta > 0$ for the VBMC [1, 2]. If $F(y) = \exp(y) - 1$ as in the MMLT [9], we have $F^{-1}(y) = \log(y + 1)$ and it can be shown that $\psi(c) = c$ for $0 < c \leq 1$; see Appendix B.2. Note that in Assumption 4, the inverse $F^{-1}$ is well-defined since $F$ is increasing and continuous.

In our analysis, we allow for the point selection procedure of ABQ itself "weak," in the sense that the optimization problem in (4) may be solved approximately.[4] That is, for a constant $0 < \tilde{\gamma} \leq 1$ we assume that the points $x_1, \ldots, x_n$ satisfy

$$a_\ell(x_{\ell+1}) \geq \tilde{\gamma} \max_{x \in \Omega} a_\ell(x), \quad (\ell = 0, 1, \ldots, n-1), \tag{9}$$

The case $\tilde{\gamma} = 1$ amounts to exactly solving the global optimization problem of ABQ (4).

The following lemma guarantees we can assume without loss of generality that the kernel matrix $K_n$ for the points $x_1, \ldots, x_n$ generated from the ABQ (4) is invertible under the assumptions above, since otherwise $\sup_{x \in \Omega} k_{X_\ell}(x, x) = 0$ holds, implying that the quadrature error is 0 from Prop. 2.1. This guarantees the applicability of Lemma 3.1 for points generated from the ABQ (4).

**Lemma 3.2.** *(proof in Appendix B.3) Suppose that Assumptions 2, 3 and 4 are satisfied. For a constant $0 < \tilde{\gamma} \leq 1$, assume that $x_1, \ldots, x_n$ are generated by a $\tilde{\gamma}$-weak version of ABQ (4), i.e., (9) is satisfied. Then either one of the following holds: i) the kernel matrix $K_\ell = (k(x_i, x_j))_{i,j=1}^\ell \in \mathbb{R}^{\ell \times \ell}$ is invertible for all $\ell = 1, \ldots, n$; or ii) there exists some $\ell = 1, \ldots, n$ such that $\sup_{x \in \Omega} k_{X_\ell}(x, x) = 0$.*

Lemma 3.1 leads to the following theorem, which establishes a connection between ABQ and weak greedy algorithms.

**Theorem 3.3.** *(proof in Appendix B.4) Suppose that Assumptions 2, 3 and 4 are satisfied. For a constant $0 < \tilde{\gamma} \leq 1$, assume that $x_1, \ldots, x_n$ are generated by a $\tilde{\gamma}$-weak version of ABQ (4), i.e., (9) is satisfied. Let $h_{x_i} = q(x_i)k(\cdot, x_i)$ for $i = 1, \ldots, n$. Then $h_{x_1}, \ldots, h_{x_n}$ are a $\gamma$-weak greedy approximation of $\mathcal{C}_{k,q}$ in $\mathcal{H}_k$ with $\gamma = \sqrt{\psi(\tilde{\gamma} C_L / C_U)}$.*

## 4 Convergence Rates of Adaptive Bayesian Quadrature

We use the connection established in the previous section to derive convergence rates of ABQ. To this end we introduce a quantity called *Kolmogorov n-width*, which is defined (for a Hilbert space $\mathcal{H}$ and a compact subset $\mathcal{C} \subset \mathcal{H}$) by

$$d_n(\mathcal{C}) := \inf_{U_n} \sup_{h \in \mathcal{C}} \text{dist}(h, U_n),$$

where the infimum is taken over all $n$-dimensional subspaces $U_n$ of $\mathcal{H}$. This is the worst case error for the best possible solution using $n$ elements in $\mathcal{H}$; thus $d_n(\mathcal{C}) \leq e_n(\mathcal{C})$ holds for any choice of $S_n$ that defines the worst case error $e_n(\mathcal{C})$ in (5). The following result by DeVore et al. [13, Corollary 3.3] relates the Kolmogorov $n$-width with the worst case error $e_n(\mathcal{C})$ of a weak greedy algorithm.

**Lemma 4.1.** *Let $\mathcal{H}$ be a Hilbert space and $\mathcal{C} \subset \mathcal{H}$ be a compact subset. For $0 < \gamma \leq 1$, let $h_1, \ldots, h_n \in \mathcal{C}$ be a $\gamma$-weak greedy approximation of $\mathcal{C}$ in $\mathcal{H}$ for $n \in \mathbb{N}$, and let $e_n(\mathcal{C})$ be the worst case error (5) for the subspace $S_n := \text{span}(h_1, \ldots, h_n)$. Then we have:*

– **Exponential decay:** *Assume that there exist constants $\alpha > 0$, $C_0 > 0$ and $D_0 > 0$ such that $d_n(\mathcal{C}) \leq C_0 \exp(-D_0 n^\alpha)$ holds for all $n \in \mathbb{N}$. Then $e_n(\mathcal{C}) \leq \sqrt{2C_0} \gamma^{-1} \exp(-D_1 n^\alpha)$ holds for all $n \in \mathbb{N}$ with $D_1 := 2^{-1-2\alpha} D_0$.*

– **Polynomial decay:** *Assume that there exist constants $\alpha > 0$ and $C_0 > 0$ such that $d_n(\mathcal{C}) \leq C_0 n^{-\alpha}$ holds for all $n \in \mathbb{N}$. Then $e_n(\mathcal{C}) \leq C_1 n^{-\alpha}$ holds for all $n \in \mathbb{N}$ with $C_1 := 2^{5\alpha+1}\gamma^{-2}C_0$.*

– **Generic case:** *We have $e_n(\mathcal{C}) \leq \sqrt{2}\gamma^{-1}\min_{1 \leq \ell < n}(d_\ell(\mathcal{C}))^{n-\ell}$ for all $n \in \mathbb{N}$. In particular, $e_{2n}(\mathcal{C}) \leq \sqrt{2}\gamma^{-1}\sqrt{d_n(\mathcal{C})}$ holds for all $n \in \mathbb{N}$.*

Thus, the key is how to upper-bound the Kolmogorov $n$-width $d_n(\mathcal{C}_{k,q})$ for the RKHS $\mathcal{H}_k$ associated with the covariance kernel $k$. Given such an upper bound, one can then derive convergence rates for ABQ using Thm. 3.3.

Below we demonstrate such results in the setting where $\Omega \subset \mathbb{R}^d$ is compact and $\mu$ is the Lebesgue measure, focusing on kernels with infinite smoothness such as Gaussian and (inverse) multiquadric kernels, using Lemma 4.1 for the case of exponential decay. In a similar way (using Lemma 4.1 for the polynomial decay case) one can also derive rates for kernels with finite smoothness, such as Matérn and Wendland kernels. These additional results are presented in Appendix C.4. We emphasize that one can also analyze other cases (e.g. kernels on a sphere) by deriving upper-bounds on the Kolmogorov $n$-width and using Thm. 3.3.

## 4.1 Convergence Rates for Kernels with Infinite Smoothness

We consider kernels with infinite smoothness, such as square-exponential kernels $k(x, x') = \exp(-\|x - x'\|^2/\gamma^2)$ with $\gamma > 0$, multiquadric kernels $k(x, x') = (-1)^{\lceil \beta \rceil}(c^2 + \|x - x'\|^2)^\beta$ with $\beta, c > 0$ such that $\beta \notin \mathbb{N}$, where $\lceil \beta \rceil$ denotes the smallest integer greater than $\beta$, and inverse multiquadric kernels $k(x, x') = (c^2 + \|x - x'\|^2)^{-\beta}$ with $\beta > 0$. We have the following bound on the Kolmogorov $n$-width of the $\mathcal{C}_{k,q}$ for these kernels; the proof is in Appendix C.2.

**Proposition 4.2.** *Let $\Omega \subset \mathbb{R}^d$ be a cube, and suppose that Assumption 2 is satisfied. Let $k$ be a square-exponential kernel or an (inverse) multiquadric kernel. Then there exist constants $C_0, D_0 > 0$ such that $d_n(\mathcal{C}_{k,q}) \leq C_0 \exp(-D_0 n^{1/d})$ holds for all $n \in \mathbb{N}$.*

The requirement for $\Omega$ to be a cube stems from the use of Wendland [38, Thm. 11.22] in our proof, which requires this condition. In fact, this can be weakened to $\Omega$ being a compact set satisfying an interior cone condition, but the resulting rate weakens to $O(\exp(-D_1 n^{-1/2d}))$ (note that this is still exponential); see [38, Sec. 11.4]. This also applies to the following results. Combining Prop. 4.2 with Lemma 3.1, Thm. 3.3 and Lemma 4.1, we now obtain a bound on $\sup_{x \in \Omega} q(x)\sqrt{k_{X_n}(x, x)}$.

**Theorem 4.3.** *(proof in Appendix C.3) Suppose that Assumptions 2, 3 and 4 are satisfied. Let $\Omega \subset \mathbb{R}^d$ be a cube, and $k$ be a square-exponential kernel or an (inverse) multiquadric kernel. For a constant $0 < \tilde{\gamma} \leq 1$, assume that $X_n = \{x_1, \ldots, x_n\} \subset \Omega$ are generated by a $\tilde{\gamma}$-weak version of ABQ (4), i.e., (9) is satisfied. Then there exist constants $C_1, D_1 > 0$ such that*

$$\sup_{x \in \Omega} q(x)\sqrt{k_{X_n}(x, x)} \leq C_1 \psi(\tilde{\gamma}C_L/C_U)^{-1/2}\exp(-D_1 n^{1/d}) \quad (n \in \mathbb{N}).$$

As a directly corollary of Prop. 2.1 and Thm. 4.3, we finally obtain a convergence rate of the ABQ with an infinitely smooth kernel, which is exponentially fast.

**Corollary 4.4.** *Suppose that Assumptions 1, 2, 3 and 4 are satisfied, and that $C_{\pi/q} := \int_\Omega \pi(x)/q(x)d\mu(x) < \infty$. Let $\Omega \subset \mathbb{R}^d$ be a cube, and $k$ be a square-exponential kernel or a (inverse) multiquadric kernel. For a constant $0 < \tilde{\gamma} \leq 1$, assume that $X_n = \{x_1, \ldots, x_n\} \subset \Omega$ are generated by a $\tilde{\gamma}$-weak version of ABQ (4), i.e., (9) is satisfied. Then there exists a constant $D_1 > 0$ independent of $n \in \mathbb{N}$ such that*

$$\left| \int f(x)\pi(x)d\mu(x) - \int T\left(m_{g,X_n}(x)\right)\pi(x)d\mu(x) \right| = O(\exp(-D_1 n^{1/d})) \quad (n \to \infty).$$

## 4.2 Discussions of the Weak Adaptivity Condition (Assumption 3)

We discuss consequences of our results to individual ABQ methods reviewed in Sec. 2.2. We do this in particular by discussing the weak adaptivity condition (Assumption 3), which requires that the data-dependent term $b_n(x)$ in (4) is uniformly bounded away from zero and infinity. (A discussion

for VBMC by Acerbi [1, 2] is given in Appendix C.8. To summarize, Assumption 3 holds if the densities of the variational distributions are bounded away uniformly from zero and infinity.)

We first consider the WSABI-L approach by Gunter et al. [16], for which $b_n(x) = (m_{g,X_n}(x))^2$; a similar result is presented for the WSABI-M in Appendix C.7. The following bounds for $b_n(x)$ follow from Lemma C.5 in Appendix C.5.

**Lemma 4.5.** *Let* $b_n(x) := (m_{g,X_n}(x))^2$. *Suppose that Assumption 1 is satisfied, and that* $\inf_{x \in \Omega} |m(x)| > 2\|\tilde{g}\|_{\mathcal{H}_k}\|k\|_{L_\infty(\Omega)}^{1/2}$. *Then Assumption 3 holds for* $C_L := \left( \inf_{x \in \Omega} |m(x)| - 2\|\tilde{g}\|_{\mathcal{H}_k}\|k\|_{L_\infty(\Omega)}^{1/2} \right)^2 > 0$ *and* $C_U := \left( \|m\|_{L_\infty(\Omega)} + 2\|\tilde{g}\|_{\mathcal{H}_k}\|k\|_{L_\infty(\Omega)}^{1/2} \right)^2 < \infty$.

Lemma 4.5 implies that WSABI-L may *not* be consistent when, e.g., one uses the zero prior mean function $m(x) = 0$, since in this case the condition $\inf_{x \in \Omega} |m(x)| > 2\|\tilde{g}\|_{\mathcal{H}_k}\|k\|_{L_\infty(\Omega)}^{1/2}$ is not satisfied. Intuitively, the inconsistency may happen because the posterior mean $m_{g,X_n}(x)$ for inputs $x$ in regions distant from the current design points $x_1, \ldots, x_n$ would become close to 0, since the prior mean function is 0; and such regions will never be explored in the subsequent iterations, because of the form $b_n(x) = (m_{g,X_n}(x))^2$. One simple way to guarantee the consistency is to make a modification like $b_n(x) := \frac{1}{2}(m_{g,X_n}(x))^2 + \alpha = T(m_{g,X_n}(x))$; then we can guarantee that $C_L \geq \alpha > 0$, encouraging exploration in the whole region $\Omega$. This then makes the algorithm consistent.

We next consider the MMLT method by Chai and Garnett [9], for which $b_n(x) = \exp(k_{X_n}(x, x) + 2m_{g,X_n}(x))$. Lemma 4.6 below shows that the weak adaptivity condition holds for the MMLT as long as Assumption 1 is satisfied. Therefore different from the WSABI, the MMLT is consistent without requiring a further assumption.

**Lemma 4.6.** *(proof in Appendix C.6) Let* $b_n(x) := \exp(k_{X_n}(x, x) + 2m_{g,X_n}(x))$. *Suppose that Assumption 1 is satisfied. Then Assumption 3 holds for* $C_L := \exp(-2\|m\|_{L_\infty(\Omega)} - 4\|\tilde{g}\|_{\mathcal{H}_k}\|k\|_{L_\infty(\Omega)}^{1/2}) > 0$ *and* $C_U := \exp(\|k\|_{L_\infty(\Omega)} + 2\|m\|_{L_\infty(\Omega)} + 4\|\tilde{g}\|_{\mathcal{H}_k}\|k\|_{L_\infty(\Omega)}^{1/2}) < 0$.

## 5  Conclusion and Outlook

Extending efficient numerical integration beyond the low-dimensional domain remains both a formidable challenge and a crucial desideratum for many areas. In machine learning, efficient numerical integration in the high-dimensional domain would be a game-changer for Bayesian learning. Developed by, and used in, the NeurIPS community, adaptive Bayesian quadrature is a promising new direction for progress in this fundamental problem class. So far, it has been hindered by the absence of theoretical guarantees.

In this work, we have provided the first known convergence guarantees for ABQ methods, by analyzing a generic form of their acquisition functions. Of central importance is the notion of weak adaptivity which, speaking vaguely, ensures that the algorithm asymptotically does not "overly focus" on some evaluations. It is conceptually related to ideas like detailed balance and ergodicity, which play a similar role for Markov Chain Monte Carlo methods (where, speaking equally vaguely, they guard against the same kind of locality) [cf. §6.5 & 6.6 in 33]. Like those of MCMC, our sufficient conditions for consistency span a flexible class of design options, and can thus act as a guideline for the design of novel acquisition functions for ABQ, guided by practical and intuitive considerations. Based on the results presented herein, novel ABQ methods may be proposed for novel domains other than only positive integrands, for example integrands with discontinuities [31] and those with spatially inhomogeneous smoothness.

An important theoretical question, however, remains to be addressed: While our results provide convergence guarantees for ABQ methods, they do not provide a theoretical explanation for why, how and when ABQ methods should be fundamentally *better* than non-adaptive methods. In fact, little is known about theoretical properties of adaptive quadrature methods in general. In applied mathematics, they remain an open problem [23, 24, 25, 26]. While we have to leave this question of ABQ's potential advantages over standard BQ for future research, we consider this area to be highly promising on account of the fundamental role of high-dimensional integrals of structured functions in probabilistic machine learning.

## Acknowledgements

We would like to express our gratitude to the anonymous reviewers for their constructive feedback. We also thank Alexandra Gessner, Hans Kersting, Tim Sullivan and George Wynne for their comments and for fruitful discussions. The authors gratefully acknowledge financial supports by the European Research Council through ERC StG Action 757275 / PANAMA, by the DFG Cluster of Excellence "Machine Learning – New Perspectives for Science", EXC 2064/1, project number 390727645, by the German Federal Ministry of Education and Research (BMBF) through the Tübingen AI Center (FKZ: 01IS18039A, 01IS18039B), and by the Ministry of Science, Research and Arts of the State of Baden-Württemberg.

## Footnotes

[2]For instance, Wenliang et al. [39, Fig. 3] used $10^{10}$ Monte Carlo samples to estimate the the normalizing constant of their model, on problems with medium dimensionality (10 to 50 dims).

[3]The point is that, in contrast to the integral over $f$, this estimate should be analytically tractable. This depends on the choices for $T$, $k$ and $\pi$. For instance, for $T(y) = y$ or $T(y) = \alpha + \frac{1}{2}y^2$ with $k$ and $\pi$ Gaussian, the estimate can be obtained analytically [16], while for $T(y) = \exp(y)$ one needs approximations; [cf. 9].

[4] We thank George Wynne for pointing out that our analysis can be extended to this weak version of ABQ.

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
