[Supplementary Material · Adaptive_Bayesian_Quadrature (25) - supp camera.pdf]

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

# A  Appendices for Section 2

## A.1  Proof of Prop. 2.1

In the proof we use the following notation: $\|\sqrt{k}\|_{L_\infty} := \sup_{x\in\Omega}\sqrt{k(x,x)}$ and $\|\sqrt{k_{X_n}}\|_{L_\infty} := \sup_{x\in\Omega}\sqrt{k_{X_n}(x,x)}$.

*Proof.* It is known that (see e.g. [18, Prop. 3.10]) the GP posterior standard deviation can be written as

$$\sqrt{k_{X_n}(x,x)} = \sup_{u\in\mathcal{H}_k:\|u\|_{\mathcal{H}_k}\leq 1}|u(x) - \boldsymbol{k}_n(x)^\top K_n^{-1}\boldsymbol{u}_n|, \quad x\in\Omega \tag{10}$$

where $\boldsymbol{u} := (u(x_1),\ldots,u(x_n))^\top \in \mathbb{R}^n$. Note that for any $x\in\Omega$, we have $m_{g,X_n}(x) = m(x) + \boldsymbol{k}_n^\top(x)K_n^{-1}\tilde{\boldsymbol{g}}_n$, since $\tilde{\boldsymbol{g}}_n = (\tilde{g}(x_i))_{i=1}^n = (m(x_i) - g(x_i))_{i=1}^n = \boldsymbol{m}_n - \boldsymbol{g}_n$. Therefore by $g(x) = m(x) + \tilde{g}(x)$, $\tilde{g}\in\mathcal{H}_k$ and (10) we have

$$|g(x) - m_{g,X_n}(x)| = |\tilde{g}(x) - \boldsymbol{k}_n^\top(x)K_n^{-1}\tilde{\boldsymbol{g}}| \leq \|\tilde{g}\|_{\mathcal{H}_k}\sqrt{k_{X_n}(x,x)}. \tag{11}$$

On the other hand, by Taylor's theorem, there exists $\alpha_{x,X_n}\in[0,1]$ such that for $y_{x,X_n} := g(x) + \alpha_{x,X_n}(m_{g,X_n}(x) - g(x)) \in \mathbb{R}$ we have

$$T(m_{g,X_n}(x)) = T(g(x)) + T'(y_{x,X_n})(m_{g,X_n}(x) - g(x)),$$

where $T'(y)$ denotes the derivative of $T$ at $y\in\mathbb{R}$. From this and (11) we have

$$|T(g(x)) - T(m_{g,X_n}(x))| \leq |T'(y_{x,X_n})||m_{g,X_n}(x) - g(x)| \leq |T'(y_{x,X_n})|\|\tilde{g}\|_{\mathcal{H}_k}\sqrt{k_{X_n}(x,x)}$$

Note that $|T'(y_{x,X_n})|$ is uniformly bounded over all $x\in\Omega$ and $n\in\mathbb{N}$, since $T'$ is continuous by assumption and $|y_{x,X_n}|$ is bound uniformly over all $x\in\Omega$ and $n\in\mathbb{N}$; the latter can be shown as

$$|y_{x,X_n}| \leq |g(x)| + |\alpha_{x,X_n}(m_{g,X_n}(x) - g(x))| \leq |m(x)| + |\tilde{g}(x)| + |m_{g,X_n}(x) - g(x)|$$

$$\leq \|m\|_{L_\infty(\Omega)} + \|\tilde{g}\|_{\mathcal{H}_k}\sqrt{k(x,x)} + \|\tilde{g}\|_{\mathcal{H}_k}\sqrt{k_{X_n}(x,x)} \leq \|m\|_{L_\infty(\Omega)} + 2\|\tilde{g}\|_{\mathcal{H}_k}\|\sqrt{k}\|_{L_\infty},$$

where we used $|\tilde{g}(x)| = |\langle\tilde{g}, k(\cdot,x)\rangle_{\mathcal{H}_k}| \leq \|\tilde{g}\|_{\mathcal{H}_k}\sqrt{k(x,x)}$ and $k_{X_n}(x,x) \leq k(x,x)$. This implies that

$$|T'(y_{x,X_n})| \leq \sup_{y\in\mathbb{R}:|y|\leq\|m\|_{L_\infty(\Omega)}+2\|\tilde{g}\|_{\mathcal{H}_k}\|\sqrt{k}\|_{L_\infty}}|T'(y)| =: C_{\tilde{g},m,k,T} < \infty.$$

Therefore,

$$|T(g(x)) - T(m_{g,X_n}(x))| \leq C_{\tilde{g},m,k,T}\|\tilde{g}\|_{\mathcal{H}_k}\sqrt{k_{X_n}(x,x)},$$

which implies that

$$\left|\int T(g(x))\pi(x)d\mu(x) - \int T(m_{g,X_n}(x))\pi(x)d\mu(x)\right|$$

$$\leq \int |T(g(x)) - T(m_{g,X_n}(x))|\pi(x)d\mu(x)$$

$$\leq C_{\tilde{g},m,k,T}\|\tilde{g}\|_{\mathcal{H}_k}\int\sqrt{k_{X_n}(x,x)}\pi(x)d\mu(x)$$

$$\leq C_{\tilde{g},m,k,T}C_{\pi/q}\|\tilde{g}\|_{\mathcal{H}_k}\sup_{x\in\Omega}q(x)\sqrt{k_{X_n}(x,x)},$$

where the last inequality follows from Hölder's inequality. $\qquad\square$

## A.2  Bound on Quadrature Error for an Alternative Estimator

We show here that for the quadrature estimator $\int\mathbb{E}_{\acute{g}}T(\acute{g}(x))\pi(x)d\mu(x)$, where $\acute{g}\sim\mathcal{GP}(m_{g,X_n}, k_{X_n})$ is the posterior Gaussian process, the essentially same upper bound as Proposition 2.1 holds, under an additional condition that

$$\mathbb{E}_{\acute{g}}\left[T'\left(|g(x)| + |\acute{g}(x)|\right)^2\right] < C, \quad \forall x\in\Omega, \forall n\in\mathbb{N}, \tag{12}$$

holds for some $C > 0$, where $T'$ is the derivative of $T$. This condition can be shown to be satisfied for transformations $T$ considered in the paper.

**Proposition A.1.** *Let $\Omega$ be a compact metric space, $X_n = \{x_1, \ldots, x_n\} \subset \Omega$ be such that the kernel matrix $K_n = (k(x_i, x_j))_{i,j=1}^n \in \mathbb{R}^{n \times n}$ is invertible, and $\pi : \Omega \to [0, \infty)$ and $q : \Omega \to [0, \infty)$ be continuous functions such that $C_{\pi/q} := \int_\Omega \pi(x)/q(x)d\mu(x) < \infty$. Suppose that Assumption 1 is satisfied. Let $\acute{g} \sim \mathcal{GP}(m_{g,X_n}, k_{X_n})$ and assume (12) is satisfied for some $C > 0$. Then we have*

$$\left| \int f(x)\pi(x)d\mu(x) - \int \mathbb{E}_{\acute{g}}T(\acute{g}(x))\pi(x)d\mu(x) \right| \leq \sqrt{2C(1 + \|\tilde{g}\|_{\mathcal{H}_k}^2)} C_{\pi/q} \sup_{x \in \Omega} q(x)\sqrt{k_{X_n}(x,x)}.$$

*Proof.* First fix $x \in \Omega$. By Taylor's theorem, there exists $\alpha_{x,X_n,\acute{g}} \in [0,1]$ such that for $y_{x,X_n,\acute{g}} := g(x) + \alpha_{x,X_n,\acute{g}}(\acute{g}(x) - g(x))$ we have

$$T(\acute{g}(x)) = T(g(x)) + T'(y_{x,X_n,\acute{g}})(\acute{g}(x) - g(x)).$$

Therefore,

$$
\begin{aligned}
(\mathbb{E}_{\acute{g}}[T(\acute{g}(x))] - T(g(x)))^2 &= (\mathbb{E}_{\acute{g}}[T'(y_{x,X_n,\acute{g}})(\acute{g}(x) - g(x))])^2 \\
&\leq \mathbb{E}_{\acute{g}}[(T'(y_{x,X_n,\acute{g}}))^2]\mathbb{E}_{\acute{g}}[(\acute{g}(x) - g(x))^2] \\
&\leq C\mathbb{E}_{\acute{g}}[(\acute{g}(x) - g(x))^2],
\end{aligned}
$$

where the last inequality follows from $|y_{x,X_n,\acute{g}}| \leq |g(x)| + |\acute{g}(x)|$ and the assumption (12). Moreover,

$$
\begin{aligned}
\mathbb{E}_{\acute{g}}[(\acute{g}(x) - g(x))^2] &\leq 2\mathbb{E}_{\acute{g}}[(\acute{g}(x) - m_{g,X_n}(x))^2] + 2(m_{g,X_n}(x) - g(x))^2 \\
&\leq 2k_{X_n}(x,x) + 2\|\tilde{g}\|_{\mathcal{H}_k}^2 k_{X_n}(x,x),
\end{aligned}
$$

where the last inequality follows from (11). Thus,

$$|T(g(x)) - \mathbb{E}_{\acute{g}}[T(\acute{g}(x))]| \leq \sqrt{2C(1 + \|\tilde{g}\|_{\mathcal{H}_k}^2)}\sqrt{k_{X_n}(x,x)}$$

and it follows that

$$
\begin{aligned}
&\left| \int T(g(x))\pi(x)d\mu(x) - \int \mathbb{E}_{\acute{g}}[T(\acute{g}(x))]\pi(x)d\mu(x) \right| \\
&\leq \int |T(g(x)) - \mathbb{E}_{\acute{g}}[T(\acute{g}(x))]|\pi(x)d\mu(x) \\
&\leq \sqrt{2C(1 + \|\tilde{g}\|_{\mathcal{H}_k}^2)} \int \sqrt{k_{X_n}(x,x)}\pi(x)d\mu(x) \\
&\leq \sqrt{2C(1 + \|\tilde{g}\|_{\mathcal{H}_k}^2)} C_{\pi/q} \sup_{x \in \Omega} q(x)\sqrt{k_{X_n}(x,x)},
\end{aligned}
$$

where the last inequality follows from Hölder's inequality.

$\square$

# B  Appendices for Section 3

## B.1  Proof of Lemma 3.1

*Proof.* It is easy to show by the reproducing property that the GP posterior variance $k_{X_n}(x,x)$ in (3) can be written as the squared RKHS distance between $k(\cdot, x)$ and its orthogonal projection onto $\text{span}(k(\cdot, x_1), \ldots, k(\cdot, x_n)) \subset \mathcal{H}_k$, provided that the kernel matrix $K_n = (k(x_i, x_j))_{i,j=1}^n \in \mathbb{R}^{n \times n}$ is invertible:

$$k_{X_n}(x,x) = \text{dist}^2(k(\cdot, x), \text{span}(k(\cdot, x_1), \ldots, k(\cdot, x_n))) = \inf_{\alpha_1,\ldots,\alpha_n \in \mathbb{R}} \left\| k(\cdot, x) - \sum_{i=1}^n \alpha_i k(\cdot, x_i) \right\|_{\mathcal{H}_k}^2.$$

Therefore,

$$
\begin{aligned}
q^2(x)k_{X_n}(x,x) &= \inf_{\alpha_1,\ldots,\alpha_n \in \mathbb{R}} \left\| q(x)k(\cdot, x) - \sum_{i=1}^n \alpha_i q(x)k(\cdot, x_i) \right\|_{\mathcal{H}_k}^2 \\
&= \inf_{\beta_1,\ldots,\beta_n \in \mathbb{R}} \left\| q(x)k(\cdot, x) - \sum_{i=1}^n \beta_i q(x_i)k(\cdot, x_i) \right\|_{\mathcal{H}_k}^2, \\
&= \inf_{g \in S_n} \|h_x - g\|_{\mathcal{H}_k}^2 = \text{dist}^2(h_x, S_n),
\end{aligned}
$$

where the second equality follows from $q(x) > 0$ and $q(x_i) > 0$ for all $i = 1, \ldots, n$; this proves (7). Using this, (5) and the definition of $\mathcal{C}_{k,q}$, the identity (8) can be shown as

$$e_n(\mathcal{C}_{k,q}) = \sup_{h \in \mathcal{C}_{k,q}} \mathrm{dist}(h, S_n) = \sup_{x \in \Omega} \mathrm{dist}(h_x, S_n) = \sup_{x \in \Omega} q(x)\sqrt{k_{X_n}(x,x)}.$$

$\square$

Fig. 2 provides a geometric interpretation of (7) in Lemma 3.1 and its proof.

Figure 2: A geometric interpretation of (7) in Lemma 3.1, for a simple case where $n = 2$. The yellow plane represents the subspace $S_2 := \mathrm{span}(k(\cdot, x_1), k(\cdot, x_2)) = \mathrm{span}(q(x_1)k(\cdot, x_1), q(x_2)k(\cdot, x_2))$, where the identity follows from $q(x_1), q(x_2) > 0$.

## B.2  An Example for Assumption 4

The following lemma gives the constant $\psi(c)$ in Assumption 4 for the case $F(y) = \exp(y) - 1$, and thus $F^{-1}(y) = \log(1 + y)$, of the MMLT [9]: $\psi(c) = c$. The proof is elementary, but we include it for completeness.

**Lemma B.1.** *For any $0 < c \leq 1$, we have $\log(1 + cy) \geq c \log(1 + y)$ for all $y \geq 0$.*

*Proof.* The assertion is equivalent to that $1 + cy \geq (1 + y)^c$ holds for all $y \geq 0$, which we show below. Let $f(y) := 1 + cy$ and $g(y) := (1 + y)^c$ for $y \geq 0$. Their derivatives are $f'(y) = c$ and $g'(y) = c(1 + y)^{c-1}$, for which we have $f'(y) \geq g'(y)$ for all $y \geq 0$, since $c - 1 \leq 0$. We also have $f(0) = g(0) = 1$. Therefore, by the fundamental theorem of calculus, we conclude that $f(y) = f(0) + \int_0^y f'(\tilde{y})d\tilde{y} \geq g(0) + \int_0^y g'(\tilde{y})d\tilde{y} = g(y)$ for all $y \geq 0$. $\square$

## B.3  Proof of Lemma 3.2

*Proof.* Let $\ell = 1, \ldots, n - 1$, and assume that $x_1, \ldots, x_\ell \in \Omega$ are such that the kernel matrix $K_\ell = (k(x_i, x_j))_{i,j=1}^\ell \in \mathbb{R}^{\ell \times \ell}$ is invertible; this is always true for $\ell = 1$. For $x_{\ell+1} \in \Omega$ such that $a_\ell(x_{\ell+1}) \geq \tilde{\gamma} \max_{x \in \Omega} a_\ell(x) = \tilde{\gamma} \max_{x \in \Omega} F\left(q^2(x)k_{X_\ell}(x,x)\right) b_\ell(x)$ with $0 < \tilde{\gamma} \leq 1$, we show that either of the following holds: i) $k(\cdot, x_{\ell+1})$ is linearly independent to $k(\cdot, x_1), \ldots, k(\cdot, x_\ell)$ and thus $K_{\ell+1} = (k(x_i, x_j))_{i,j=1}^{\ell+1} \in \mathbb{R}^{(\ell+1) \times (\ell+1)}$ is invertible, or ii) $\sup_{x \in \Omega} k_{X_\ell}(x, x) = 0$.

Assume that ii) does not hold. Then there exists $y \in \Omega$ such that $k_{X_\ell}(y, y) > 0$. For this $y$ we have $a_\ell(y) = F\left(q^2(y)k_{X_\ell}(x,x)\right) b_\ell(x) > 0$, since $q(x), b_\ell(x) > 0$ for all $x \in \Omega$, $F(0) = 0$ and $F$ is increasing. Therefore $a_\ell(x_{\ell+1}) \geq \tilde{\gamma} a_\ell(y) > 0$, and thus $k_{X_\ell}(x_{\ell+1}, x_{\ell+1}) > 0$. Note that since the

kernel matrix $K_\ell$ is invertible, we have

$$k_{X_\ell}(x_{\ell+1}, x_{\ell+1}) = \inf_{\alpha_1,\ldots,\alpha_n \in \mathbb{R}} \left\| k(\cdot, x_{\ell+1}) - \sum_{i=1}^{\ell} \alpha_i k(\cdot, x_i) \right\|_{\mathcal{H}_k}^2$$

This expression and $k_{X_\ell}(x_{\ell+1}, x_{\ell+1}) > 0$ imply that $k(\cdot, x_{\ell+1})$ is linearly independent to $k(\cdot, x_1), \ldots, k(\cdot, x_\ell)$, since otherwise $k(\cdot, x_{\ell+1})$ can be written as a linear combination of $k(\cdot, x_1), \ldots, k(\cdot, x_\ell)$, and thus $k_{X_\ell}(x_{\ell+1}, x_{\ell+1})$ becomes 0 from the above expression. Thus i) has been shown. □

### B.4 Proof of Theorem 3.3

*Proof.* For $\ell = 0, \ldots, n-1$, by $a_\ell(x_{\ell+1}) \geq \tilde{\gamma} \sup_{x\in\Omega} a_\ell(x)$ and Assumption 3, we have

$$
\begin{aligned}
a_\ell(x_{\ell+1}) &\geq \tilde{\gamma} \sup_{x\in\Omega} F\left(q^2(x)k_{X_\ell}(x,x)\right) b_\ell(x) \\
&\geq \tilde{\gamma}C_L \sup_{x\in\Omega} F\left(q^2(x)k_{X_\ell}(x,x)\right) = \tilde{\gamma}C_L F\left(\sup_{x\in\Omega} q^2(x)k_{X_\ell}(x,x)\right),
\end{aligned}
$$

where the last equality follows from $F$ being an increasing function. This implies by Assumption 3 that

$$F\left(q^2(x_{\ell+1})k_{X_\ell}(x_{\ell+1}, x_{\ell+1})\right) \geq (\tilde{\gamma}C_L/C_U)F\left(\sup_{x\in\Omega} q^2(x)k_{X_\ell}(x,x)\right)$$

and therefore, again by $F$ being increasing and also by Assumption 4,

$$
\begin{aligned}
q^2(x_{\ell+1})k_{X_\ell}(x_{\ell+1}, x_{\ell+1}) &\geq F^{-1}\left((\tilde{\gamma}C_L/C_U)F\left(\sup_{x\in\Omega} q^2(x)k_{X_\ell}(x,x)\right)\right) \\
&\geq \psi(\tilde{\gamma}C_L/C_U) \sup_{x\in\Omega} q^2(x)k_{X_\ell}(x,x).
\end{aligned}
$$

Note that $\|h_x\|_{\mathcal{H}_k}^2 = \|q(x)k(\cdot,x)\|_{\mathcal{H}_k}^2 = q^2(x)k(x,x)$ for all $x \in \Omega$. Therefore for $\ell = 0$, in which case $k_{X_0}(x,x) = k(x,x)$, we have $\|h_{x_1}\|_{\mathcal{H}_k} \geq \sqrt{\psi(\tilde{\gamma}C_L/C_U)} \sup_{x\in\Omega} \|h_x\|_{\mathcal{H}_k} = \sqrt{\psi(\tilde{\gamma}C_L/C_U)} \sup_{h\in\mathcal{C}_{k,q}} \|h\|_{\mathcal{H}_k}$. For $\ell = 1, \ldots, n-1$ we have by Lemma 3.1 (which is applicable from Assumption 2),

$$
\begin{aligned}
\mathrm{dist}^2(h_{x_{\ell+1}}, S_\ell) &= q^2(x_{\ell+1})k_{X_\ell}(x_{\ell+1}, x_{\ell+1}) \\
&\geq \psi(\tilde{\gamma}C_L/C_U) \sup_{x\in\Omega} \mathrm{dist}^2(h_x, S_\ell) = \psi(\tilde{\gamma}C_L/C_U) \sup_{h\in\mathcal{C}_{k,q}} \mathrm{dist}^2(h, S_\ell).
\end{aligned}
$$

Thus (6) holds for $\gamma = \sqrt{\psi(\tilde{\gamma}C_L/C_U)}$, which completes the proof. □

## C  Appendices for Section 4

### C.1  A Bound on the Kolmogorov n-width

**Lemma C.1.** *Let $x_1, \ldots, x_n \in \Omega$ be such that the kernel matrix $K_n = (k(x_i, x_j))_{i,j=1}^n \in \mathbb{R}^{n\times n}$ is invertible. Assume that $q(x) > 0$ for all $x \in \Omega$. Then we have $d_n(\mathcal{C}_{k,q}) \leq \inf_{x_1,\ldots,x_n\in\Omega} \sup_{x\in\Omega} q(x)\sqrt{k_{X_n}(x,x)}$.*

*Proof.* Using Lemma 3.1, the Kolmogorov $n$-width can be upper-bounded as

$$
\begin{aligned}
d_n(\mathcal{C}_{k,q}) &= \inf_{U_n} \sup_{h\in\mathcal{C}_{k,q}} \mathrm{dist}(h, U_n) = \inf_{U_n} \sup_{x\in\Omega} \mathrm{dist}(h_x, U_n) \\
&\leq \inf_{x_1,\ldots,x_n\in\Omega} \sup_{x\in\Omega} \mathrm{dist}(h_x, S_n) = \inf_{x_1,\ldots,x_n\in\Omega} \sup_{x\in\Omega} q(x)\sqrt{k_{X_n}(x,x)},
\end{aligned}
$$

where the infimum in the first line is taken over all $n$-dimensional subspaces $U_n$ of $\mathcal{H}_k$, and $S_n = \mathrm{span}(h_{x_1}, \ldots, h_{x_n})$ with $h_x = q(x)k(\cdot, x)$. □

Lemma C.1 can be used for deriving upper-bounds on the Kolmogorov $n$-width $d_n(\mathcal{C}_{k,q})$ for concrete examples of the kernel $k$ on $\Omega \subset \mathbb{R}^d$. To this end, the key quantity is the *fill distance* defined by

$$h_{X_n,\Omega} := \sup_{x \in \Omega} \min_{i=1,\ldots,n} \|x - x_i\|,$$

where $X_n := \{x_1, \ldots, x_n\} \subset \Omega$. This measures how densely the points $x_1, \ldots, x_n$ fill the region $\Omega$.

## C.2 Proof of Prop. 4.2 (Kolmogorov n-width for kernels with infinite smoothness)

*Proof.* By [38, Theorem 11.22], where $k_{X_n}(x,x)$ is called the *power function*, there is a constant $c > 0$ such that $k_{X_n}(x,x) \le \exp(-c_1/h_{X_n,\Omega})$ holds for any set of design points $X_n = \{x_1, \ldots, x_n\}$ with sufficiently small $h_{X_n,\Omega}$. If we define $x_1, \ldots, x_n$ as equally-spaced grid points in $\Omega$, then we have $h_{X_n,\Omega} = c_2 n^{-1/d}$ for some $c_2 > 0$ independent of $n$. Therefore for large enough $n$, we have $k_{X_n}(x,x) \le \exp(-c_1 c_2^{-1} n^{1/d})$. In other words, there exists $n_0 \in \mathbb{N}$ such that $k_{X_n}(x,x) \le \exp(-(c_1/c_2)n^{1/d})$ holds for all $n \ge n_0$. Note that there exists a constant $c_3 > 0$ such that $k_{X_n}(x,x') \le c_3$ holds for all $x \in \Omega$ and for all $n$, since $\Omega$ is compact and $k_{X_n}(x,x)$ is continuous w.r.t. $x$ for any fixed $n$ and non-increasing w.r.t. $n$ for any fixed $x \in \Omega$.

Now, define $c_4 > 0$ as a constant such that $c_4 \exp(-(c_1/c_2)n_0^{1/d}) = c_3$, and let $c_5 := \max(c_4, 1)$. Then, for $n < n_0$ we have $c_5 \exp(-(c_1/c_2)n^{1/d}) \ge c_4 \exp(-(c_1/c_2)n^{1/d}) \ge c_3 \ge k_{X_n}(x,x)$. For $n \ge n_0$, we have $c_5 \exp(-(c_1/c_2)n^{1/d}) \ge \exp(-(c_1/c_2)n^{1/d}) \ge k_{X_n}(x,x)$. Therefore we conclude that $k_{X_n}(x,x) \le c_5 \exp(-(c_1/c_2)n^{1/d})$ holds for all $n \in \mathbb{N}$ and $x \in \Omega$.

Note that $\inf_{x_1,\ldots,x_n \in \Omega} \sup_{x \in \Omega} q(x)\sqrt{k_{X_n}(x,x)} \le \sup_{x \in \Omega} q(x)\sqrt{k_{X_n}(x,x)}$ holds for any fixed choice of $x_1, \ldots, x_n$ defining $k_{X_n}(x,x)$ in the upper-bound. If we chose $x_1, \ldots, x_n$ as equally-spaced grid points in the upper-bound, we have that $d_n(\mathcal{C}_{k,q}) \le \inf_{x_1,\ldots,x_n \in \Omega} \sup_{x \in \Omega} q(x)\sqrt{k_{X_n}(x,x)} \le \sup_{x \in \Omega} q(x)\sqrt{c_5} \exp(-\frac{1}{2}c_1 c_2^{-1} n^{1/d})$ by Lemma C.1 and the above argument. Setting $C_0 := \sup_{x \in \Omega} q(x)\sqrt{c_5}$ and $D_0 := \frac{1}{2}c_1 c_2^{-1}$ concludes the proof. $\square$

## C.3 Proof of Theorem 4.3

*Proof.* By Thm. 3.3, $h_{x_1}, \ldots, h_{x_n}$ are a $\gamma$-weak approximation of $\mathcal{C}_{k,q}$ in $\mathcal{H}_k$ with $\gamma = \sqrt{\psi(\tilde{\gamma}C_L/C_U)}$. From this, and by Lemma 4.1 (exponential) and Prop. 4.2, there exist $C_0, D_0 > 0$ such that for $C_1 := \sqrt{2C_0}$ and $D_1 := 2^{-1-2/d}D_0$, we have $e_n(\mathcal{C}_{k,q}) \le C_1 \psi(\tilde{\gamma}C_L/C_U)^{-1/2} \exp(-D_1 n^{-1/d})$ for all $n \in \mathbb{N}$. Combining this and (8) in Lemma 3.1 concludes the proof. $\square$

## C.4 Convergence Rates for ABQ using Kernels with Finite Smoothness

We deal with here kernels with finite smoothness. In particular, we consider shift-invariant kernels of the form $k(x,x') = \Phi(x - x')$ with $\Phi \in L_1(\mathbb{R}^d)$ satisfying

$$c_1(1 + \|\omega\|^2)^{-r} \le \hat{\Phi}(\omega) \le c_2(1 + \|\omega\|^2)^{-r}, \quad \omega \in \mathbb{R}^d \tag{13}$$

for some $c_1, c_2 > 0$ and $r > d/2$, where $\hat{\Phi}$ denotes the Fourier transform of $\Phi$. The RKHS of such a kernel is norm-equivalent to a Sobolev space of order $r$, which consists of functions whose weak derivative up to order $r$ exist and are square-integrable [38, Corollary 10.48]; thus $r$ represents the smoothness of functions in the RKHS.

For instance, Matérn kernels [32, p. 84] of the form

$$k(x,x') = \frac{2^{1-\nu}}{\Gamma(\nu)} \left( \frac{\sqrt{2\nu}\|x - x'\|}{\ell} \right)^\nu K_\nu \left( \frac{\sqrt{2\nu}\|x - x'\|}{\ell} \right), \quad (\nu, \ell > 0)$$

where $\Gamma$ is the Gamma function and $K_\nu$ is the modified Bessel function of second kind, satisfy (13) with $r = \nu + d/2$. Another example is Wendland kernels [38, Theorem 10.35], which have compact supports and thus have computational advantages; see [37] and [38, Chapter 9] for details. In the following result, we use the notion of a Lipschitz boundary and an interior cone condition, the definitions of which can be found in, e.g., [20, Section 3] and references therein.

**Assumption 5.** $\Omega \subset \mathbb{R}^n$ *is a compact set having a Lipshitz boundary and satisfying an interior cone condition.*

### C.4.1 Kolmogorov n-width for kernels with finite smoothness

**Proposition C.2.** *Suppose that Assumptions 2 and 5 are satisfied. Let $k(x, x') = \Phi(x - x')$ be a kernel satisfying* (13) *for $r > d/2$. Then there exists a constant $C_0 > 0$ such that*

$$d_n(\mathcal{C}_{k,q}) \leq C_0 n^{-r/d+1/2}, \quad n \in \mathbb{N}.$$

*Proof.* By [38, Corollary 11.33] (where we set $m = 0$ and $q = \infty$), there exists a constant $c_1 > 0$ such that for all $g \in \mathcal{H}_k$ we have

$$\|g - m_{g,X_n}\|_{L_\infty(\Omega)} \leq c_1 h_{X_n,\Omega}^{r-d/2} \|g\|_{\mathcal{H}_k},$$

for $X_n = \{x_1, \ldots, x_n\} \subset \Omega$ with sufficiently small $h_{X_n,\Omega}$. By setting $x_1, \ldots, x_n$ as equally-spaced grid points in $\Omega$, there exists a constant $c_2 > 0$ such that $h_{X_n,\Omega} \leq c_2 n^{-1/d}$. Therefore we have for some $c_3 > 0$

$$\sup_{g \in \mathcal{H}_k : \|g\|_{\mathcal{H}_k} \leq 1} \|g - m_{g,X_n}\|_{L_\infty(\Omega)} \leq c_3 n^{-r/d+1/2}$$

for sufficiently large $n$. Note that the GP posterior variance can be written as (see e.g. [18, Prop. 3.10])

$$\sqrt{k_{X_n}(x,x)} = \sup_{g \in \mathcal{H}_k : \|g\|_{\mathcal{H}_k} \leq 1} |g(x) - m_{g,X_n}(x)|, \quad x \in \Omega.$$

This implies that $\sqrt{k_{X_n}(x,x)} \leq \sup_{\|g\|_{\mathcal{H}_k} \leq 1} \|g - m_{g,X_n}\|_{L_\infty(\Omega)}$ for all $x \in \Omega$, which further implies that $\sup_{x \in \Omega} \sqrt{k_{X_n}(x,x)} \leq \sup_{\|g\|_{\mathcal{H}_k} \leq 1} \|g - m_{g,X_n}\|_{L_\infty(\Omega)}$. Therefore, for large enough $n$ we have $\sup_{x \in \Omega} \sqrt{k_{X_n}(x,x)} \leq c_3 n^{-r/d+1/2}$ if $x_1, \ldots, x_n$ are equally-spaced grid points in $\Omega$. In other words, there exists $n_0 \in \mathbb{N}$ such that

$$\sup_{x \in \Omega} \sqrt{k_{X_n}(x,x)} \leq c_3 n^{-r/d+1/2}, \quad \forall n \geq n_0.$$

Note that there exists a constant $c_4 > 0$ such that $\sqrt{k_{X_n}(x,x)} \leq c_4$ holds for all $x \in \Omega$ and for all $n \in \mathbb{N}$, since $\Omega$ is compact, $k_{X_n}(x,x)$ is continuous w.r.t. $x$ for any fixed $n$ and $k_{X_n}(x,x)$ is non-increasing w.r.t. $n$ for any fixed $x \in \Omega$. Therefore $\sup_{x \in \Omega} \sqrt{k_{X_n}(x,x)} \leq c_4$ for all $n \in \mathbb{N}$.

Now, define $c_5 > 0$ as a constant such that $c_5 n_0^{-r/d+1/2} = c_4$, and let $c_6 := \max(c_5, c_3)$. Then, for $n < n_0$ we have $c_6 n^{-r/d+1/2} \geq c_5 n^{-r/d+1/2} \geq c_4 \geq \sup_{x \in \Omega} \sqrt{k_{X_n}(x,x)}$. For $n \geq n_0$, we have $c_6 n^{-r/d+1/2} \geq c_3 n^{-r/d+1/2} \geq \sup_{x \in \Omega} \sqrt{k_{X_n}(x,x)}$. Therefore we conclude that, if $x_1, \ldots, x_n$ are equally-spaced grid points in $\Omega$, we have

$$\sup_{x \in \Omega} \sqrt{k_{X_n}(x,x)} \leq c_6 n^{-r/d+1/2}, \quad \forall n \in \mathbb{N},$$

Finally, by Lemma C.1 we have

$$d_n(\mathcal{C}_{k,q}) \leq \sup_{x \in \Omega} q(x) c_6 n^{-r/d+1/2}$$

and thus the assertion holds with $C_0 := \sup_{x \in \Omega} q(x) c_6 < \infty$, which is bounded since $q$ is continuous and $\Omega$ is compact.

$\square$

### C.4.2 Convergence Rates

Combining Prop. C.2 and Thm. 3.3, we have the following bound on $\sup_{x \in \Omega} q(x) \sqrt{k_{X_n}(x,x)}$, for $x_1, \ldots, x_n$ are generated by a $\tilde{\gamma}$-weak version of ABQ (9) with a constant $0 < \tilde{\gamma} \leq 1$.

**Theorem C.3.** *Suppose that Assumptions 2, 3, 4 and 5 are satisfied. Let $k(x, x') = \Phi(x - x')$ be a kernel satisfying* (13) *for $r > d/2$. For a constant $0 < \tilde{\gamma} \leq 1$, assume that $X_n = \{x_1, \ldots, x_n\} \subset \Omega$ are generated by a $\tilde{\gamma}$-weak version of ABQ* (4), *i.e.,* (9) *is satisfied. Then there exists a constant $C_1 > 0$ such that*

$$\sup_{x \in \Omega} q(x) \sqrt{k_{X_n}(x,x)} \leq C_1 \psi(\tilde{\gamma} C_L/C_U)^{-1} n^{-r/d+1/2}, \quad n \in \mathbb{N}.$$

*Proof.* Let $h_x := q(x)k(\cdot, x)$ for any $x \in \Omega$. Then by Thm. 3.3, $h_{x_1}, \ldots, h_{x_n}$ are a $\gamma$-weak greedy approximation of $\mathcal{C}_{k,q}$ in $\mathcal{H}_k$ with $\gamma = \sqrt{\psi(\tilde{\gamma}C_L/C_U)}$. From this, and by Lemma 4.1 (polynomial decay) and Prop. C.2, there exists a constant $C_0 > 0$ such that $e_n(\mathcal{C}_{k,q}) \leq 2^{5\alpha+2}\gamma^{-2}C_0 n^{-\alpha}$ holds for all $n \in \mathbb{N}$, where $\alpha := r/d - 1/2$. Combining this inequality and (8) yields assertion with $C_1 = 2^{5\alpha+2}C_0$. $\qquad\square$

As a corollary of Prop. 2.1 and Thm. C.3, we have the following result.

**Corollary C.4.** *Suppose that Assumptions 1, 2, 3, 4 and 5 are satisfied, and that $C_{\pi/q} := \int |\pi(x)/q(x)|d\mu(x) < \infty$. Assume $k(x, x') = \Phi(x - x')$ satisfies (13) with $r > d/2$. For a constant $0 < \tilde{\gamma} \leq 1$, assume that $X_n = \{x_1, \ldots, x_n\} \subset \Omega$ are generated by a $\tilde{\gamma}$-weak version of ABQ (4), i.e., (9) is satisfied. Then we have*

$$\left| \int f(x)\pi(x)d\mu(x) - \int T(m_{g,X_n}(x))\pi(x)d\mu(x) \right| = O(n^{-r/d+1/2}) \quad (n \to \infty).$$

## C.5 Bounds for GP Posterior Mean Functions

The following lemma is used for deriving the constants $C_L$ and $C_U$ in Assumption 3 for individual ABQ methods.

**Lemma C.5.** *Assume that $\tilde{g} := g - m \in \mathcal{H}_k$. Then for all $x \in \Omega$ and $n \in \mathbb{N}$, we have*

$$|m(x)| - 2\|\tilde{g}\|_{\mathcal{H}_k}\sqrt{k(x,x)} \leq |m_{g,X_n}(x)| \leq |m(x)| + 2\|\tilde{g}\|_{\mathcal{H}_k}\sqrt{k(x,x)}$$

*Proof.* We show the lower-bound; the upper-bound can be shown similarly. Since $\tilde{g} \in \mathcal{H}_k$, we have

$$
\begin{aligned}
|m_{\tilde{g},X_n}(x)| &\leq |\tilde{g}(x)| + |\tilde{g}(x) - m_{\tilde{g},X_n}(x)| \\
&\leq \|\tilde{g}\|_{\mathcal{H}_k}\sqrt{k(x,x)} + \|\tilde{g}\|_{\mathcal{H}_k}\sqrt{k_{X_n}(x,x)} \leq 2\|\tilde{g}\|_{\mathcal{H}_k}\sqrt{k(x,x)}. \quad (14)
\end{aligned}
$$

Note that $m_{g,X_n}(x) = m(x) + m_{\tilde{g},X_n}(x)$ since $\tilde{g} = g - m$. Therefore,

$$|m_{g,X_n}(x)| \geq |m(x)| - |m_{\tilde{g},X_n}(x)| \geq |m(x)| - 2\|\tilde{g}\|_{\mathcal{H}_k}\sqrt{k(x,x)}.$$

$\qquad\square$

## C.6 Proof of Lemma 4.6

*Proof.* First note that $0 \leq k_{X_n}(x,x) \leq k(x,x)$ for all $x \in \Omega$ and $n \in \mathbb{N}$. Using Lemma C.5, we have

$$
\begin{aligned}
\exp(k_{X_n}(x,x) + 2m_{g,X_n}(x)) &\leq \exp(k_{X_n}(x,x) + 2|m_{g,X_n}(x)|) \\
&\leq \exp(k(x,x) + 2|m(x)| + 4\|\tilde{g}\|_{\mathcal{H}_k}\sqrt{k(x,x)}) \\
&\leq \exp(\|k\|_{L_\infty(\Omega)} + 2\|m\|_{L_\infty(\Omega)} + 4\|\tilde{g}\|_{\mathcal{H}_k}\|k\|_{L_\infty(\Omega)}^{1/2})
\end{aligned}
$$

Similarly, we have

$$
\begin{aligned}
\exp(k_{X_n}(x,x) + 2m_{g,X_n}(x)) &\geq \exp(2m_{g,X_n}(x)) \\
&\geq \exp(-2|m_{g,X_n}(x)|) \\
&\geq \exp(-2|m(x)| - 4\|\tilde{g}\|_{\mathcal{H}_k}\sqrt{k(x,x)}) \\
&\geq \exp(-2\|m\|_{L_\infty(\Omega)} - 4\|\tilde{g}\|_{\mathcal{H}_k}\|k\|_{L_\infty(\Omega)}^{1/2})
\end{aligned}
$$

$\qquad\square$

## C.7 Bounds for WSABI-M

The following bounds for $b_{X_n}(x) = \frac{1}{2}k_{X_n}(x,x) + (m_{g,X}(x))^2$ of the WSABI-M [16] can be easily obtained using Lemma C.5 and $0 \leq \frac{1}{2}k_{X_n}(x,x) \leq \frac{1}{2}\|k\|_{L_\infty(\Omega)}$.

**Lemma C.6.** *Let $b_{X_n}(x) = \frac{1}{2}k_{X_n}(x,x) + (m_{g,X}(x))^2$. Suppose that Assumption 1 is satisfied, and that $\inf_{x \in \Omega}|m(x)| > 2\|\tilde{g}\|_{\mathcal{H}_k}\|k\|_{L_\infty(\Omega)}^{1/2}$. Then $C_L < b_n(x) < C_U$ for all $x \in \Omega$ and $n \in \mathbb{N}$, where $C_L := (\inf_{x \in \Omega}|m(x)| - 2\|\tilde{g}\|_{\mathcal{H}_k}\|k\|_{L_\infty(\Omega)}^{1/2})^2 > 0$ and $C_U := \frac{1}{2}\|k\|_{L_\infty(\Omega)} + (\|m\|_{L_\infty(\Omega)} + \|k\|_{L_\infty(\Omega)}^{1/2})^2 < \infty$.*

## C.8 Discussion for Variational Bayesian Monte Carlo (VBMC)

The VBMC by Acerbi [1, 2] uses $F(y) = y^{\delta_1}$, $q(x) = 1$ and $b_n(x) = \pi_n^{\delta_2}(x)\exp(\delta_3 m_{g,X_n}(x))$, where $\pi_n$ is the variational posterior at the $n$-th iteration and $\delta_1, \delta_2, \delta_3 \geq 0$ are constants. Recall that in this method the transformation is identity: $T(y) = y$ for $y \in \mathbb{R}$; thus $g = f$. The following result can be easily obtained from Lemma C.5.

**Lemma C.7.** *Let $b_n(x) = \pi_n^{\delta_2}(x)\exp(\delta_3 m_{g,X_n}(x))$ with $\delta_2, \delta_3 \geq 0$. Suppose that Assumption 1 is satisfied, and that there exist constants $D_L, D_U$ such that $0 < D_L < \pi_n(x) < D_U < \infty$ holds for all $x \in \Omega$ and $n \in \mathbb{N}$. Then we have $C_L < b_n(x) < C_U < \infty$ for all $x \in \Omega$ and $n \in \mathbb{N}$, where $C_L := D_L^{\delta_2}\exp(-\delta_3(\|m\|_{L_\infty(\Omega)} + 2\|\tilde{g}\|_{\mathcal{H}_k}\|k\|_{L_\infty(\Omega)}^{1/2})) > 0$ and $C_U := D_U^{\delta_2}\exp(\delta_3(\|m\|_{L_\infty(\Omega)} + 2\|\tilde{g}\|_{\mathcal{H}_k}\|k\|_{L_\infty(\Omega)}^{1/2})) < \infty$.*

The condition $0 < D_L < \pi_n(x) < D_U < \infty$ for all $x \in \Omega$ and $n \in \mathbb{N}$ requires that 1) the supports of the variational distributions should cover the whole domain $\Omega$; and that ii) the density values of the variational distributions should be uniformly bounded from above. This implies that, if the variational family is a set of Gaussian mixtures (as proposed by Acerbi [1, 2]), then the variance of each mixture component should be uniformly lower- and upper-bounded; otherwise the condition $0 < D_L < \pi_n(x) < D_U < \infty$ may not be satisifed.

We note that in the setting of the VBMC, the density $\pi$ in the target integral $\int f(x)\pi(x)d\mu(x)$ is an intractable posterior density, and it is to be approximated as $\int m_{f,X_n}(x)\pi_n(x)d\mu(x)$ using the variational posterior $\pi_n$; therefore there is also an error due to the approximation of $\pi$ by $\pi_n$. Thus, a complete theoretical analysis requires analyzing the convergence behavior of the variational posterior $\pi_n$; this is out of scope of this paper and we leave it for future research.