[Reviews · NeurIPS 2019]

Reviewer 1



# After rebuttal Thank you very much for the response. I will try to further elaborate my comments below. Proposition 4.2 This result is actually the key to unlocking the contributions of the paper and not Lemma 3.2. Apologies if my review was not clear on this. I have checked the updated proof and your arguments are mathematically correct. However, I still have problems with the constants. In particular, a minor issue is that for n >= n_0 and cases with c_5 > 1 the bound is not tight. A problematic part is the case with n < n_0 and c_4 > 1. Observe that c_4 = c_3 exp ((c_1 / c_2) n_0^{1\d}) and that in the latter case k_{X_n}(x, x) <= c_3 exp ((c_1 / c_2) (n_0^{1\d} - n^{1/d})) with n_0^{1\d} - n^{1/d} > 0, which implies that exp ((c_1 / c_2) (n_0^{1\d} - n^{1/d})) > 1. The bound is worse than c_3 alone which holds for all x, irrespective of n. Thus, the updated bound does not really guarantee that for n < n_0 one will observe decay of exp (-n^{1/d}). In my understanding, the result only guarantees _asymptotic_ decay of exp(-n^{1/d}). Lemma 3.2 In my view, this result follows from Lemma 3.1 directly. Observe that Lemma 3.1 tells you that k_{X_l}(x, x) is the squared distance from the evaluation functional k(x, *) in RKHS to its projection onto the span of the evaluation functionals centered at instances from X_l (I am omitting q(x) because that can be considered as an importance weight and does not change anything). If all the evaluation functionals define a linearly independent set of vectors the kernel matrix is non-singular (n instances with n >= l). If the set is linearly dependent then there exists an evaluation functional (i.e., vector) in the span of others. For that particular vector, the distance to its projection onto the span will be zero. So, Lemma 3.2 says the set of evaluation functionals is either linearly independent or linearly dependent. # The paper studies convergence properties of adaptive Bayesian quadrature methods. This class of quadrature methods deals with an integrand for which it is known beforehand that it satisfies a certain constraint such as being positive. The main idea is to introduce a link function and apply it to the output of a latent function such that the resulting composition of the two functions satisfies the desired constraint. The latent function is modeled with a Gaussian process and the so called design points are selected deterministically by optimizing an acquisition function. The paper considers a generic acquisition function that covers several of the previously proposed strategies for selecting design points in the context of adaptive Bayesian quadrature methods. The first theoretical result provides an upper bound on the approximation error of adaptive Bayesian quadrature methods in terms of posterior variance under fairly standard assumptions (Proposition 2.1). Following this, the upper bound based on the posterior variance is related to [1] where a weak greedy algorithm is considered with an asymptotic convergence guarantee. The greedy algorithm [1] restricted to a Hilbert space deals with an approximation of the inner product on that space using the inner product defined on a subspace spanned by some landmarks. The landmarks are selected iteratively so that they are at least \gamma-fraction of the optimal choice. The quality of the approximation is measured by first projecting the original space to the subspace spanned by the landmarks and then taking the distance between a point from the original space that is furthest away from the span. This link from Proposition 2.1 to the weak greedy algorithm from [1] hinges on Lemmas 3.1 and 3.2. These two lemmas are fairly well-known results and follow directly from [2, 3] (the results can also be found in some derivations of the Nyström method). In my opinion, these two lemmas are not entirely novel and should be credited to previous work. Theorem 3.3 gives an asymptotic convergence guarantee for adaptive Bayesian quadrature methods. For the purpose of deriving convergence rates, the paper introduces the notion of Kolmogorov n-width (Section 4). In essence, it is the worst case approximation error for the best possible choice of the landmarks. Denoting the Kolmogorov n-width with d_n and the weak greedy approximation error with e_n, the following inequality can be established d_n <= e_n. Given that d_n <= e_n, it is not entirely clear from the paper itself how an upper bound on d_n implies an upper bound on e_n (Lemma 4.1). It would be nice to provide a clarification of this claim in any future version of the paper (appending part (i) from [1, Corollary 3.3]). Section 4.1 goes beyond asymptotic convergence and provides explicit convergence rates for adaptive Bayesian quadrature methods. I have some concerns regarding the proofs from this section and will try to clarify them below. # Section 4.1 (Convergence rates) & Appendix C I fail to see why the proof of Proposition 4.2 holds and will try to clarify my concerns (Appendix C, lines 480--488). i) k_{X_n} (x, x) <= exp (-c_1 / h_{X_n, \Omega}) holds for some constant c_1 and sufficiently small h_{X_n, \Omega} ii) h_{X_n, \Omega} = c_2 n^{-1 / d} holds for some constant c_2 and {x_i} selected so that they are equally-spaced grid points of \Omega iii) (i) and (ii) ==> k_{X_n} (x, x) <= exp (- (c_1 / c_2) n^{1 / d} ) for large enough n iv) there exists a constant c_3 such that k_{X_n} (x, x) <= c_3 for all n and for all x \in \Omega ----- ??? ----- v) k_{X_n} (x, x) <= c_3 exp (- (c_1 / c_2) n^{1 / d} ) for all n and for all x subject to {x_i} being equally spaced. ----- ??? ----- I can follow the reasoning in (i)-(iii) and have no problems with any of the claims in (i)-(iv). However, I cannot see why the upper bound in (v) would hold for all n. In particular, we have that (iii) holds only for large enough n which can be so large that we will never select that many points in adaptive Bayesian quadrature methods. The claim in (iv) holds for all n and for all x but as n gets larger the term exp (- (c_1 / c_2) n^{1 / d} ) << 1 and, thus, the right-hand side of (v) can fail the upper bound from (iv) for some n. Such n might also not be sufficiently large for (iii) to hold. Now, this is an important result and failure to demonstrate a convergence rate in Proposition 4.2 implies that Theorem 4.3 does not hold either. This further implies that Corollary 4.4 does not hold and there are no explicit convergence rates for adaptive Bayesian quadrature methods. Please provide clarification for this in the rebuttal, as a mistake would take away the claimed contribution. I think there is a similar problem in Appendix C.4.1 with polynomial decay rates. References: [1] DeVore, Petrova, Wojtaszczyk (Constructive approximation, 2013). Greedy algorithms for reduced bases in Banach spaces. [2] Smola & Schölkopf (ICML 2000). Sparse greedy matrix approximation for machine learning [3] Schölkopf & Smola (2000). Learning with kernels (Section 10.2, Sparse greedy matrix approximation).

Reviewer 2



Summary: The paper’s main contribution, in my opinion, is to connect a rather wide class of adaptive Bayesian quadrature algorithms to the general class of week-greedy algorithms in an RKHS, enabling the authors to analyze the convergence of these algorithms as special cases of existing theory. The paper is rather dense, and I wish the authors provided more intuition about the role and interpretation of various terms in Eqn. (4). By contrast, I think the connection to week-greedy algorithms is presented more clearly, and in my opinion is generally interesting, even for those not interested in convergence rates, just seeking to gain a different perspective on this class of algorithms. I am not aware of any published work studying convergence rates of this wide class of algorithms, and while I can’t confidently comment on how tight or 'surprising' the results are, I think it is an important first contribution in this domain. Detailed comments: Line 120: I am puzzled as to why this estimator is used for the integral. For each value of the function g, there is an estimator for the integral. However, one has to then take an expectation over the posterior over g. Here, it seems that we’re taking an expectation over g first (to obtain \mu_{g,X_n}) and then plugging the mean into the estimator. Due to Jensen’s inequality, depending on the convexity of T, this is likely to under- or overestimate the posterior mean. I briefly looked at [8] which is referenced here, but even there, Table 1 suggests a different estimator should be used. Can the authors please comment on this, and perhaps expand on this in the paper. Section 2.2: A generic form of acquisition functions is given here, but it is unclear if this form, and its various components have an intuitive interpretation. While it’s possible to check if all the referenced acquisition functions do indeed fit this general formulation, it feels very generic, and it’s unclear why this is the right level of abstraction. I would have appreciated a discussion of the role the various components in Eqn (4) play. Similarly, it is unclear to me if the authors think that any arbitrary special case of (4) has a clear interpretation as ABQ. Lines 160-167: I do wonder whether this compressed generic description of RKHS basics is useful for the reader here, I think this could be sacrificed in favour of providing more intuitive explanation of the results. Section 3: I liked this relatively compact but informative overview of gamma-week greedy algorithms.

Reviewer 3



Summary The authors analyse adaptive Bayesian quadrature in a general form. Via relating the algorithm to weak greedy set approximation algorithms in RKHS [11], they prove consistency and provide high level convergence rates of the estimated integrals. An excellent paper: highly relevant and worthwhile problem, a neat solution, very nice to read, theoretically thorough. I enjoyed reading it and have very few points of critique. Significance Adaptive Bayesian quadrature methods have been shown to work very well in practice, with many published methods. Phrasing those within a unified framework, and proving consistency and its conditions therefore is a major step forward in this field. Clarity The paper is very well written, with great introduction and conclusions. Despite the very technical nature, the series of arguments that lead to the final rates are reasonably easy to follow. Moving all proofs to the Appendix is obviously required to to space constraints, although I liked the geometric interpretation of the proof of Lemma 3.1 and think it might be a nice fit for the main text. I am not sure it is possible though If anything, the conclusions and intro are slightly redundant and potentially could be shorted in favour of the figure. However, since the paper is incomplete without the proofs anyways, maybe that is not worth it. The relation to weak greedy algorithms My compliment to the authors for realising and establishing the connection -- very neat! Simulations The paper does not contain any simulations, but I think that is justified. Bayesian quadrature vs Monte Carlo I am somewhat sympathic that the authors promote BQ over MC methods in the intro, since this is a BQ paper. However, I think the presentation is slightly one-sided: while BQ methods certainly have their merits for certain types of problems, the are definitely not a silver bullet. One obvious (and probably often invoked) comment here is that the convergence rate when estimating integrals with MC is independent of the dimension, whereas the BQ rates suffer quite a bit. Of course all this depends on the problem at hand and the context, but my point is simply that MC, especially when combined with MCMC, has been a hugely successful technique for a reason -- and it is unlikely that it will be fully replaced by BQ methods. The example cited in via [24] is not representative whatsoever, but rather cherry picking. I feel the general tone should be rephrased to be more neutral -- highlighting pros and cons of each side. Ergodicity / detailed balance / weak adaptivity The abstract, intro, and conclusions use the these terms in a quite bold fashion. However, the main text and analysis falls short in even mentioning them -- not to speak of any formal connection. I think this is bad practice (over-selling) and suggest a more minimal comment in a discussion section instead. Minor: I think an intuitive discussion of the weak adaptivity condition would be helpful immediately after stating it. Currently, this is done in the conclusion (line 316), which might be a bit late. Supplementary material / proofs The main text has a very nice flow of arguments, while the Appendix is slightly less so. I was able to more or less follow most proofs in the Appendix, but it would be helpful if the authors would provide some high level overview or map of the different components, how they interact, and what techniques are used within. This would help the reader substantially in my opinion. Thanks to the authors for commenting on my points of critique in the rebuttal. I am happy with the response and the planned edits. During the reviewer discussion, some points of critique on the proof (and the updated version from the rebuttal) were raised. Namely that the provided rate of decay only holds asymptotic (see comments of R1). It might be possible to fix that and we encourage the authors to either do that or adjust their discussion of the rate. In the discussion, we agreed that consistency is nevertheless a major step forward.

[Author Response · NeurIPS 2019]

We thank the three anonymous reviewers for their insightful and largely positive comments!

**Responses to Reviewer 1** Thanks again for your thoughtful comments.

**Lemmas 3.1 and 3.2.** Regarding Lemma 3.1, the novelty is the existence of the function $q(x)$. If $q(x) = 1$, this is well
known in the literature, as the reviewer pointed out. We will make this point clear in the revision. Regarding Lemma
3.2, while we do not see this as a key contribution of the paper, it is not clear to us whether it follows from the results of
Smola and Schoelkopf. This is because they consider greedy selection of a subset of training data, while we consider
selection of points from the entire domain. If you know any existing work that explicitly states Lemmas 3.2, could you
please let us know when you update your review? We will cite it and remove our proof in Appendix.

**Lemma 4.1.** Thank you for the suggestion, which we follow in the revision.

**Proof of Proposition 4.2.** Thank you very much for pointing this out. We agree that our argument of deriving (v) was
flawed. We make the following correction, which we believe makes Proposition 4.2 still valid. (We will correct the proof
of Prop. C.2 in a similar way.) We start from (iii), which can be stated as that there exists $n_0 \in \mathbb{N}$ such that $k_{X_n}(x, x) \leq$
$\exp(-(c_1/c_2)n^{1/d})$ holds for all $n \geq n_0$. Now, define $c_4 > 0$ as a constant such that $c_4 \exp(-(c_1/c_2)n_0^{1/d}) = c_3$, and
let $c_5 := \max(c_4, 1)$. Then, for $n < n_0$ we have $c_5 \exp(-(c_1/c_2)n^{1/d}) \geq c_4 \exp(-(c_1/c_2)n^{1/d}) \geq c_3 \geq k_{X_n}(x, x)$,
where the last inequality follows from (iv). For $n \geq n_0$, we have $c_5 \exp(-(c_1/c_2)n^{1/d}) \geq \exp(-(c_1/c_2)n^{1/d}) \geq$
$k_{X_n}(x, x)$. Therefore we conclude that $k_{X_n}(x, x) \leq c_5 \exp(-(c_1/c_2)n^{1/d})$ holds for all $n \in \mathbb{N}$ and $x \in \Omega$.

**Responses to Reviewer 2** Thanks again for your insightful comments.

**The estimator in line 120.** This is a very good point, and thanks for pointing it out. We agree that the current
presentation is confusing, and will make a correction. The quadrature estimator suggested in [8] (and used for WSABI-
M [13]) can be described as $\int \mathbb{E}_{\acute{g}} T(\acute{g}(x))\pi(x)$, where $\acute{g} \sim \mathcal{GP}(m_{g,X_n}, k_{X_n})$ is the posterior GP. On the other hand, the
estimator in line 120 is $\int T(m_{g,X_n}(x))\pi(x)$ and used by WSABI-L [13]. As we describe below, these two estimators
are *both consistent with the same convergence rates*, and all theoretical guarantees obtained in the paper are applicable
to the estimator $\int \mathbb{E}_{\acute{g}} T(\acute{g}(x))\pi(x)$ as well. Intuitively, this is because the posterior $\acute{g} \sim \mathcal{GP}(m_{g,X_n}, k_{X_n})$ contracts
around the posterior mean $m_{g,X_n}$ as $n$ increases, and $\mathbb{E}_{\acute{g}} T(\acute{g})$ and $T(m_{g,X_n})$ get similar. We note however that for a
finite $n$, it is not clear which estimator is "better," and this is an interesting topic for future research.

We sketch here that for the estimator $\int \mathbb{E}_{\acute{g}} T(\acute{g}(x))\pi(x)$, the essentially same upper bound as Proposition 2.1 holds, under
an additional condition that $\mathbb{E}_{\acute{g}}(T'(|g(x)| + |\acute{g}(x)|))^2 < C$ holds for all $x \in \Omega$ and $n \in \mathbb{N}$ for some $C > 0$ (which can be
shown to be satisfied for transformations $T$ mentioned in our paper). By Taylor's theorem, there exists $\alpha_{x,X_n,\acute{g}} \in [0, 1]$
such that for $y_{x,X_n,\acute{g}} := g(x) + \alpha_{x,X_n,\acute{g}}(\acute{g}(x) - g(x))$ we have $T(\acute{g}(x)) = T(g(x)) + T'(y_{x,X_n,\acute{g}})(\acute{g}(x) - g(x))$.
Therefore $(\mathbb{E}_{\acute{g}}[T(\acute{g}(x))] - T(g(x)))^2 = (\mathbb{E}_{\acute{g}}[T'(y_{x,X_n,\acute{g}})(\acute{g}(x) - g(x))])^2 \leq \mathbb{E}_{\acute{g}}[(T'(y_{x,X_n,\acute{g}}))^2]\mathbb{E}_{\acute{g}}[(\acute{g}(x) - g(x))^2] \leq$
$C\mathbb{E}_{\acute{g}}[(\acute{g}(x) - g(x))^2]$, where the last inequality follows from $|y_{x,X_n,\acute{g}}| \leq |g(x)| + |\acute{g}(x)|$ and the above assumption.
Moreover, $\mathbb{E}_{\acute{g}}[(\acute{g}(x) - g(x))^2] \leq 2\mathbb{E}_{\acute{g}}[(\acute{g}(x) - m_{g,X_n}(x))^2] + 2(m_{g,X_n}(x) - g(x))^2 \leq 2k_{X_n}(x, x) + 2\|\tilde{g}\|^2_{\mathcal{H}_k} k_{X_n}(x, x)$,
where the last inequality follows from Eq. (10). Thus, $|T(g(x)) - \mathbb{E}_{\acute{g}}[T(\acute{g}(x))]| \leq \sqrt{2C(1 + \|\tilde{g}\|^2_{\mathcal{H}_k})}\sqrt{k_{X_n}(x, x)}$.
Following the argument in line 436, the essentially same bound as Prop. 2.1 can be obtained (with a different constant).

**The generic form of acquisition functions (Eq. 4).** We defined Eq. (4) so that the class of acquisition functions to
which our convergence guarantees are applicable becomes as large as possible. One positive side of this generality
is that it enables practitioners to design a new acquisition that results in a consistent algorithm; its derivation can be
quite different from those of the existing acquisition functions (which are often done by approximating an intractable
posterior covariance function of the transformed integrand). In this sense, we think that every special case of (4) does
not have to have an interpretation as an approximate posterior covariance. We nevertheless agree that Eq. 4 is rather
abstract. We will include a discussion about the roles of the components in Eq. (4).

**Responses to Reviewer 3** Thanks again for encouraging comments and insightful suggestions.

**BQ v.s. MC.** We will make the tone of the comparison more neutral, including a discussion of the dependence of the
dimensionality. We hope that you agree, however, that the existence of a convergence guarantee for MCMC (which
didn't exist for adaptive BQ so far) is a key contributor to MCMC's popularity.

**Ergodicity / detailed balance and the relation to weak adaptivity.** We will modify the presentation so that the
intuitive discussion on the connection to the detailed balance and ergodicity becomes minimal. We will also add an
intuitive explanation about the weak adaptivity condition immediately after it is introduced.

**Proof map / Appendix.** We will provide a diagram in Appendix that describes the relationships between the various
auxiliary results and how they yield the main results. We will also add a high level overview of the proof plan and
techniques used.

[Meta-Review · NeurIPS 2019]

This work potentially provides a solid foundation for future non-asymptotic analyses of ABQ. The connection between adaptive Bayesian quadrature (ABQ) algorithms and certain greedy algorithms in Hilbert spaces as well as the ABQ (asymptotic) error bounds could have significant impact.